# MLL5 suppresses antiviral innate immune response by facilitating STUB1-mediated RIG-I degradation

Peipei Zhou[1], Xiaodan Ding[1], Xiaoling Wan[1], Lulu Liu[1,2], Xiujie Yuan[1], Wei Zhang[1,3], Xinhui Hui[1,3], Guangxun Meng[1], Hui Xiao[1], Bin Li[4], Jin Zhong[1], Fajian Hou[5], Lihwen Deng[6] & Yan Zhang [1]

Trithorax group protein MLL5 is an important epigenetic modifier that controls cell cycle progression, chromatin architecture maintenance, and hematopoiesis. However, whether MLL5 has a role in innate antiviral immunity is largely unknown. Here we show that MLL5 suppresses the RIG-I-mediated anti-viral immune response. *Mll5*-deficient mice infected with vesicular stomatitis virus show enhanced anti-viral innate immunity, reduced morbidity, and viral load. Mechanistically, a fraction of MLL5 located in the cytoplasm interacts with both RIG-I and its E3 ubiquitin ligase STUB1, which promotes K48-linked polyubiquitination and proteasomal degradation of RIG-I. MLL5 deficiency attenuates the RIG-I and STUB1 association, reducing K48-linked polyubiquitination and accumulation of RIG-I protein in cells. Upon virus infection, nuclear MLL5 protein translocates from the nucleus to the cytoplasm inducing STUB1-mediated degradation of RIG-I. Our study uncovers a previously unrecognized role for MLL5 in antiviral innate immune responses and suggests a new target for controlling viral infection.

[1] Key Laboratory of Molecular Virology and Immunology, Institut Pasteur of Shanghai, Chinese Academy of Sciences, University of Chinese Academy of Sciences, 320 Yueyang Road, Shanghai 200031, China. [2] Institute of Biology and Medical Sciences, Soochow University, 199 Ren'ai Road, Suzhou 215123, China. [3] School of Life Sciences, Shanghai University, 99 Shangda Road, Shanghai 200444, China. [4] Shanghai Institute of Immunology, Shanghai Jiaotong University, 280 South Chongqing Road, Shanghai 200025, China. [5] Institute of Biochemistry and Cell Biology, Chinese Academy of Sciences, 320 Yueyang Road, Shanghai 200031, China. [6] Department of Biochemistry, Yong Loo Lin School of Medicine, National University Health System, National University of Singapore, 8 Medical Drive, MD 7 #04-06, Singapore 117597, Singapore. Correspondence and requests for materials should be addressed to Y.Z. (email: yan_zhang@sibs.ac.cn)

The innate immune system provides the first line of defense against foreign pathogens. Invading pathogens trigger evolutionarily conserved pattern recognition receptors (PRRs), including Toll-like receptors (TLRs), RIG-I-like receptors (RLRs), Nod-like receptors (NLRs) and cytosolic sensors of double-stranded DNA. These PRRs initiate signaling cascades to induce production of type I interferons (IFNs) and proinflammatory cytokines and invoke the intrinsic antiviral defenses and adaptive immunity to achieve immune clearance[1]. RLR family proteins such as RIG-I and melanoma differentiation associated protein 5 (MDA5) recognize cytosolic viral RNA[2]. RIG-I recognizes 5′-triphosphate RNA (5′-pppRNA) as well as short double-stranded RNA (dsRNA), while MDA5 recognizes longer dsRNA. RIG-I consists of two N-terminal caspase recruitment domains (CARD), a central helicase domain and a C-terminal RD domain (CTD). In an inactive state, the CARD domain of RIG-I binds to its central helicase domain. After binding to viral RNA, cytosolic RIG-I undergoes conformational changes, oligomerization, and exposes its CARD domains for association with the N-terminal CARD domain of mitochondrial antiviral-signaling protein (MAVS) at the mitochondrial outer membrane. The heterotypic CARD–CARD interaction between RIG-I and MAVS then recruits tumor necrosis factor (TNF) receptor (TNFR)-associated factor 3 (TRAF3) to activates TANK-binding kinase (TBK)1/IKKε signaling or IKKα/IKKβ/NEMO [nuclear factor (NF)-κB essential modulator] signaling and activates transcription factors IFN-regulatory factor (IRF)3 and NF-κB. Activated IRF3 and NF-κB then translocate into the nucleus and induce production of type I IFNs and proinflammatory cytokines[2].

Several post-translational modifications, especially the ubiquitination of RIG-I, are important for regulation of RIG-I-mediated immune signaling. K63-linked polyubiquitination of RIG-I by E3 ligases such as Riplet, tripartite motif-containing protein (Trim) 25, Trim4, or MEX3C is required for RIG-I activation by altering conformation[3–7]. Conversely, deubiquitinating enzymes such as CYLD, ubiquitin-specific protease (USP)3, and USP21 inhibit RIG-I-mediated immune signaling by removing the K63 ubiquitin chain from RIG-I protein[8–10]. In contrast, E3 ubiquitin ligases such as ring finger protein (RNF)122, RNF125, or STUB1 regulate RIG-I protein stability by inducing K48-linked polyubiquitination and proteasome-dependent degradation[11–13]. RIG-I is a type-I-IFN-induced protein and inappropriate production of inflammatory cytokines leads to immune toxicity, therefore, degradation of RIG-I illustrates an important mechanism for attenuation of RNA-virus-induced immune responses.

Mixed lineage leukemia (MLL)5 belongs to the evolutionarily conserved Trithorax family, which induces histone H3K4 transferase activity and contributes to a broad range of biological processes[14]. Previous studies have demonstrated that Mll5 deficiency in mice results in impairment of self-renewal ability of hematopoietic stem cells (HSCs)[15–17], and accumulation of DNA damage and reactive oxygen species (ROS) in HSCs[18]. It has been shown that MLL5 favors the self-renewal of giloblastoma by repressing histone variant H3.3[19]. An isoform of MLL5 that is specifically expressed in HPV16/18-positive cervical cancer cells is essential for transcriptional activation of the E6/E7 oncogene[20]. Another isoform of MLL5 protein, NKp44L, is a ligand for natural killer (NK) cell NKp44 receptor that mediates natural cytotoxicity toward tumor cells[21]. Nevertheless, whether MLL5 plays a role in innate antiviral immunity is largely unknown.

In the present study, we show that MLL5 acts as a negative regulator in host antiviral immune responses. A fraction of MLL5 that was located in the cytoplasm and mediated interaction between RIG-I and its E3 ubiquitin ligase STUB1, leads to K48-linked polyubiquitination and proteasomal degradation of RIG-I. Ablation of MLL5 attenuated interaction between RIG-I and STUB1, and reduced K48-linked polyubiquitination and accumulation of RIG-I protein in cells. MLL5 deficiency potentiates the production of type I IFN, proinflammatory cytokines and innate antiviral immune responses to RNA virus both in vitro and in vivo. Moreover, upon viral infection, MLL5 protein translocates from the nucleus to the cytoplasm to induce STUB1-mediated RIG-I degradation. Here we show an unexpected role for MLL5 in host antiviral immune responses, highlighting a mechanism of epigenetic modifiers in controlling viral infection.

## Results

**MLL5 suppresses RLR-mediated innate immune responses**. To explore the function of MLL5 in the antiviral immune response, we generated Mll5 deficient (Mll5$^{-/-}$) mice using CRISPR/Cas9 approach (Supplementary Fig. 1a), which caused 8 nucleotides deletion in exon 3 (Supplementary Fig. 1b and c). Mll5-deficient mice newly generated in this study exhibited perinatal lethality, postnatal growth retardation, impaired male fertility, and compromised hematopoietic reconstitution ability (Supplementary Fig. 1d-f), which is consistent with previous reports[15–17].

We next generated bone marrow-derived macrophages (BMDMs) from wild-type or Mll5$^{-/-}$ mice, and challenged them with diverse pathogen-associated molecular pattern (PAMP) ligands. The mRNA expression of type I IFN and proinflammatory cytokines were detected using quantitative reverse transcription PCR (qRT-PCR). We found that Mll5$^{-/-}$ BMDMs expressed upregulated Ifn-β, Tnf-α, and Il-6 mRNA compared with those from wild-type BMDMs after synthetic RNA duplex poly(I:C) (polyinosinic:polycytidylic acid) or 5′-pppRNA transfection, but not stimulation with other PAMP ligands, such as lipopolysaccharide (LPS) (TLR4 ligand), CpG-B (TLR9 ligand), R848 (TLR7/8 ligand), Pam3 (TLR1/2 ligand), poly(I:C)(TLR3 ligand), or intracellular IFN stimulatory DNA (ISD) (Fig. 1a). To test this further, we prepared primary peritoneal macrophages (PMs) or mouse embryonic fibroblasts (MEFs) from wild-type or Mll5$^{-/-}$ mice, and transfected them with poly(I:C) or 5′-pppRNA. In line with that, the levels of Ifn-β and Tnf-α or Il-6 mRNA and the production of IFN-β and TNF-α or IL-6 cytokines were significantly higher in Mll5$^{-/-}$ PMs (Fig. 1b, c) than in wild-type cells when transfected with poly(I:C) or 5′-pppRNA, but not intracellular ISD.

Cytosolic viral RNA, transfected poly(I:C) and 5′-pppRNA are recognized in cells by RLRs. To determine whether MLL5 affected RNA-virus-induced antiviral immune responses, we infected wild-type or Mll5$^{-/-}$ PMs with vesicular stomatitis virus (VSV) or Sendai virus (SeV), then measured mRNA expression and cytokine production of IFN-β and TNF-α or IL-6. The DNA virus herpes simplex virus type 1 (HSV-1) was used as a negative control. We found that Mll5$^{-/-}$ PMs had higher gene expression and protein secretion of IFN-β, TNF-α, and IL-6 than their wild-type counterparts had in response to infection with VSV or SeV, but not HSV-1 (Fig. 1b, c). Similar results were observed in Mll5$^{-/-}$ MEF cells treated with poly(I:C) transfection (Supplementary Fig. 2a, b) or VSV infection (Supplementary Fig. 2c, d).

We next generated MLL5$^{-/-}$ HEK293T human embryonic kidney cells using a CRISPR-Cas9-based approach, and detected the role of MLL5 in antiviral immune responses in human cells (Supplementary Fig. 3a). Similarly, MLL5$^{-/-}$ HEK293T cells increased intracellular poly(I:C)-induced expression of IFN-β and TNF-α (Supplementary Fig. 3b), indicating that the function of MLL5 in the antiviral immune response is conserved in mice and humans. Therefore, these results demonstrated that MLL5 selectively suppresses RLR-mediated production of type I IFN and proinflammatory cytokines.

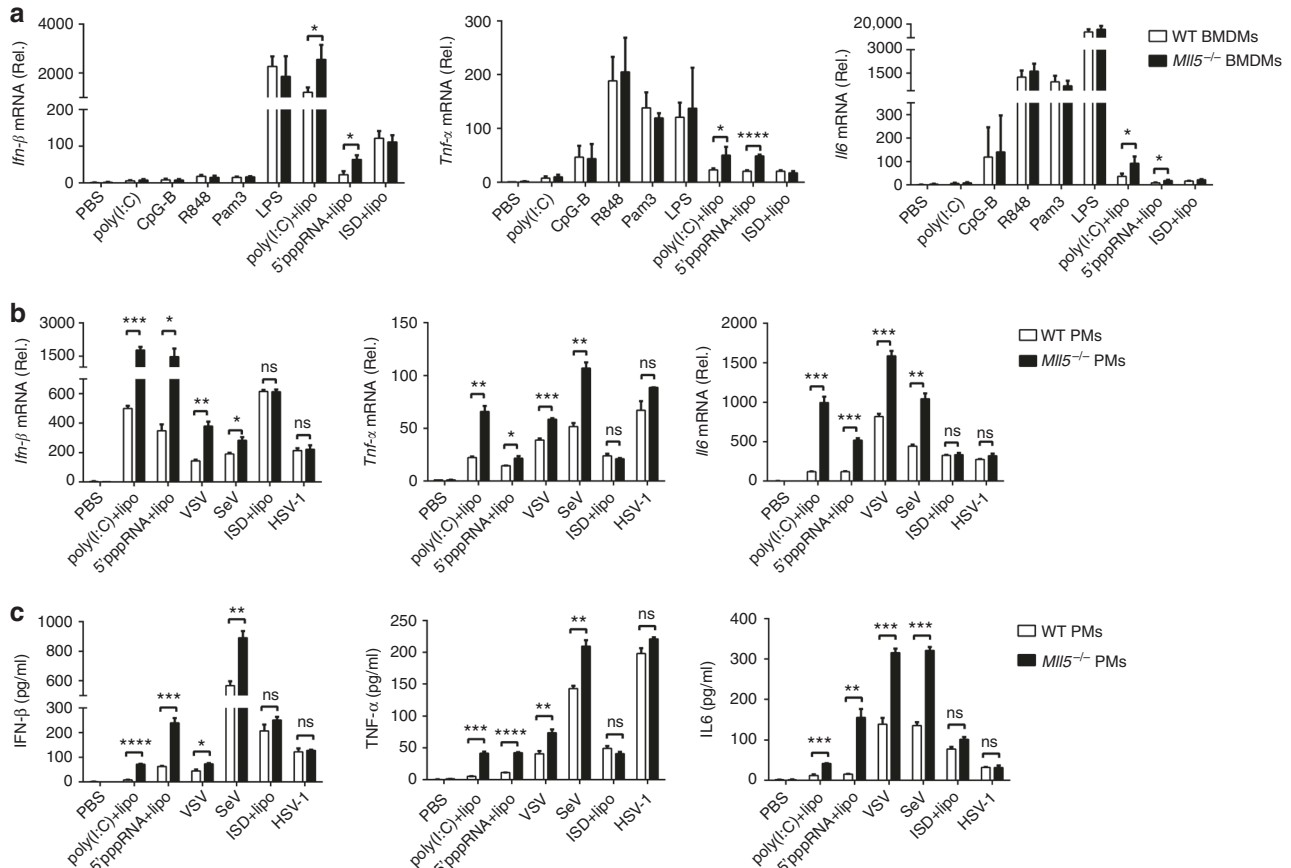

**Fig. 1** MLL5 selectively suppresses RLR-mediated antiviral immune response. **a** Expression of *Ifn-β*, *Tnf-α*, and *Il6* mRNA in BMDMs from wild-type (WT) or *Mll5⁻/⁻* mice stimulated with poly(I:C) (100 μg/ml), CpG-B (1 μg/ml), R848 (1 μg/ml), Pam3 (1 μg/ml) and LPS (0.2 μg/ml) for 4 h, or stimulated with intracellular poly(I:C) (1 μg/ml), intracellular 5′ppp-RNA (0.4 μg/ml) and intracellular ISD (1 μg/ml) for 6 h. *Gapdh* served as control. **b** Expression of *Ifn-β*, *Tnf-α* and *Il6* mRNA in PMs from WT or *Mll5⁻/⁻* mice stimulated with intracellular poly(I:C) (1 μg/ml), intracellular 5′ppp-RNA (0.4 μg/ml) and intracellular ISD (1 μg/ml) for 6 h, or infected with VSV-GFP (MOI:1), SeV (10 HA/ml) and HSV-1 (MOI:1) for 6 h. *Gapdh* served as control. **c** ELISA quantification of IFN-β, TNF-α and IL6 secretion in PMs treated as in **b**. Data were from three independent experiments and were analyzed by Student's *t*-test (two-tailed) and were presented as mean ± SD (**p* < 0.05, ***p* < 0.01,****p* < 0.001, *****p* < 0.0001, ns means no significant difference)

**MLL5 suppresses RLR-mediated antiviral signaling**. To investigate further the effect of MLL5 in RLR-mediated immune signaling, we detected activation of transcription factors IRF3 and NF-κB by immunoblot assays in wild-type or *Mll5⁻/⁻* MEFs infected with VSV. Phosphorylation levels of IRF3 and NF-κB subunit P65 were higher in *Mll5⁻/⁻* than in wild-type MEFs in response to VSV infection (Fig. 2a). After activation, phosphorylated IRF3 and P65 translocate from the cytoplasm to the nucleus to activate transcription of type I IFNs and proinflammatory cytokines. We therefore examined whether the nuclear translocation of IRF3 and P65 in response to VSV infection was affected by MLL5 deficiency. Consistent with increased phosphorylation of IRF3 and P65, the levels of IRF3 and P65 nuclear translocation were significantly increased in *Mll5⁻/⁻* MEFs compared with wild-type MEFs after VSV infection (Fig. 2b). Similarly, phosphorylation levels of IRF3 and P65 were higher in *MLL5⁻/⁻* HEK293T cells than in wild-type cells in response to VSV infection (Supplementary Fig. 3c).

We next performed an overexpression experiment to elucidate the effect of MLL5 in RLR-mediated immune signaling. MLL5-expressing plasmids were cotransfected into HEK293T cells with IFN-β, NF-κB, or IFN-stimulated response element (ISRE) luciferase reporter plasmids, followed by transfection with poly(I:C). Ectopic overexpression of MLL5 protein markedly inhibited intracellular poly(I:C)-induced IFN-β, NF-κB and ISRE promoter-driven luciferase activities in both wild-type and *MLL5⁻/⁻* HEK293T cells in a dose-dependent manner (Fig. 2c). Collectively, these results demonstrated that MLL5 suppresses RLR-triggered immune signaling pathways.

**MLL5 deficiency inhibits RNA virus replication in vivo**. To confirm the role of MLL5 in antiviral innate immune responses during RNA virus infection, we infected wild-type or *MLL5⁻/⁻* HEK293T cells with VSV expressing green fluorescent protein (GFP), and monitored VSV replication by flow cytometry analysis of GFP-positive cells. Flow cytometry showed that VSV replication was substantially inhibited in *MLL5⁻/⁻* HEK293T cells, suggesting that *MLL5* deficiency suppresses RNA virus replication in vitro (Fig. 3a).

To elucidate the function of MLL5 in host defense against RNA virus infection in vivo, we challenged wild-type or *Mll5⁻/⁻* mice intravenously with the VSV. The level of IFN-β, TNF-α, and IL-6 was significantly higher in the sera of *Mll5⁻/⁻* mice than in wild-type mice after VSV infection (Fig. 3b). Expression of *Ifn-β*, *Tnf-α*, and *Il-6* mRNA in the liver and lung tissues was significantly higher in *Mll5⁻/⁻* than in wild-type mice (Fig. 3c, d). Consistent with that, titer and replication of VSV were suppressed in *Mll5⁻/⁻* mice relative to those in wild-type mice (Fig. 3e, f). Hematoxylin and eosin staining of the lungs after VSV infection

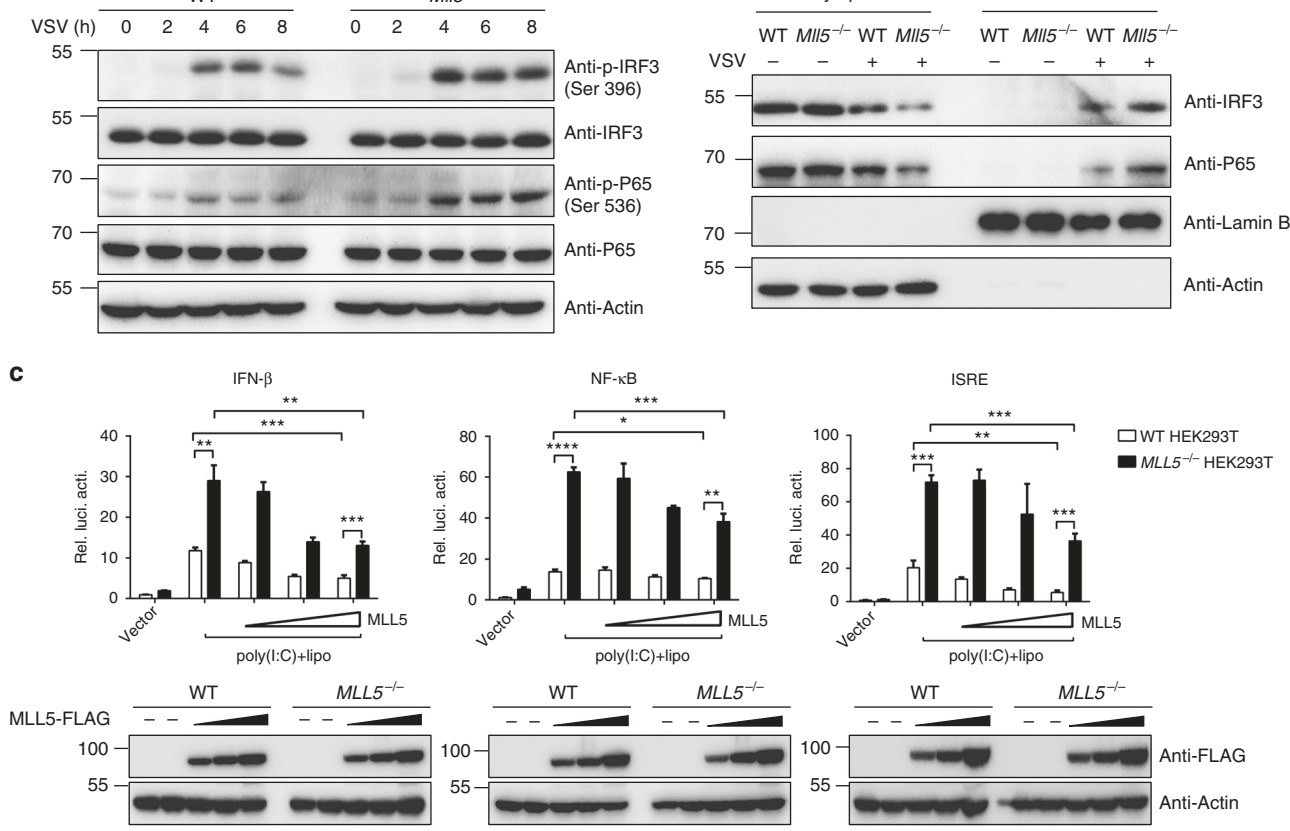

**Fig. 2** MLL5 suppresses RLR-mediated immune signaling. **a** Immunoblot analysis of phosphorylated (p-) IRF3 and P65 in WT and $Mll5^{-/-}$ MEFs upon infection with VSV-GFP (MOI:1). Actin served as a loading control. **b** Immunoblot analysis of IRF3 and P65 protein in nuclear and cytoplasmic fractions in WT and $Mll5^{-/-}$ MEFs infected with VSV-GFP (MOI:1). Actin served as a cytoplasmic control. Lamin B1 served as a nuclear protein control. **c** WT and $MLL5^{-/-}$ HEK293T cells were transiently transfected with indicated reporter plasmids (200 ng) along with MLL5-expressing plasmids (0, 100, 300, and 600 ng). After 24 h, a luciferase assay was performed with stimulation of intracellular poly(I:C) (1 μg/ml) for 16 h. Results were presented relative to the luciferase activity in control cells (transfected with luciferase reporter and empty vector without stimulation of intracellular poly(I:C)). Immunoblot analysis of MLL5-FLAG are shown below. Actin served as a loading control. Data were representative of three independent experiments with similar results (**a**, **b**) or were from three independent experiments (**c**), and were analyzed by Student's t-test (two-tailed) and were presented as mean ± SD (*$p < 0.05$, **$p < 0.01$, ***$p < 0.001$, ****$p < 0.0001$)

showed less infiltration of inflammatory cells and less injury in $Mll5^{-/-}$ mice than in wild-type mice (Fig. 3g). Compared to wild-type mice, $Mll5^{-/-}$ mice exhibited less body weight loss and regained the body weight more quickly upon low-dose VSV infection (Fig. 3h), and had significantly less mortality than wild-type mice upon high-dose VSV infection (Fig. 3i). These results indicated that $Mll5^{-/-}$ mice are more resistant to VSV infection than wild-type mice are, further demonstrating a negative role of MLL5 in innate immune responses against RNA viruses.

**MLL5 deficiency results in accumulation of RIG-I protein**. The RLR family consists of three members: RIG-I, MDA5, and LGP2. RIG-I and MDA5 but not LGP2 recognize cytosolic RNA and are recruited to mitochondrial MAVS, and then trigger downstream antiviral immune signaling cascades, eventually leading to the activation of NF-κB and IRF3. To determine the mechanism by which MLL5 inhibits RLR signaling pathway, we cotransfected MLL5-expressing vector and ISRE promoter-driven luciferase reporter plasmid in HEK293T cells together with MAVS-, TBK1- or IRF3- expressing vectors. MLL5 overexpression did not affect ISRE reporter activation induced by MAVS, TBK1 or IRF3, indicating that MLL5 represses RLR signaling by interfering with an upstream effector (Fig. 4a).

We next examined the levels of critical proteins in the RLR-mediated signaling pathway in wild-type and $MLL5^{-/-}$ HEK293T cells. The levels of MDA5 and MAVS were comparable between wild-type and $MLL5^{-/-}$ HEK293T cells. By contrast, we found a dramatic increase of RIG-I protein levels in $MLL5^{-/-}$ HEK293T cells (Fig. 4b). Similarly, the level of RIG-I protein was higher in $Mll5^{-/-}$ than in wild-type BMDMs, suggesting that $Mll5$ deficiency-induced accumulation of RIG-I is conserved in mice and humans (Fig. 4c). We used a CRISPR-Cas9-based approach to insert sequences encoding the 3×FLAG epitope immediately before the stop codon of the endogenous $DDX58$ gene (encodes RIG-I protein) in wild-type or $MLL5^{-/-}$ HEK293T cell lines (Supplementary Fig. 4a, b). Similarly, the levels of FLAG-tagged RIG-I protein were significantly increased in $DDX58$ 3×FLAG knock-in $MLL5^{-/-}$ HEK293T cell lines compared with $DDX58$ 3×FLAG knock-in wild-type HEK293T cells by immunoblot using an anti-FLAG antibody (Supplementary Fig. 4c).

To investigate further whether MLL5 deficiency-augmented innate immune response is dependent on RIG-I, we knocked down RIG-I through sh-RNAs in wide-type and $Mll5^{-/-}$ MEFs, and challenged them with VSV and HSV-1, followed by detection of IFN-β and TNF-α expression using qRT-PCR (Fig. 4d). In line with previous results, expression of $Ifn$-$β$ and $Tnf$-$α$ mRNA was

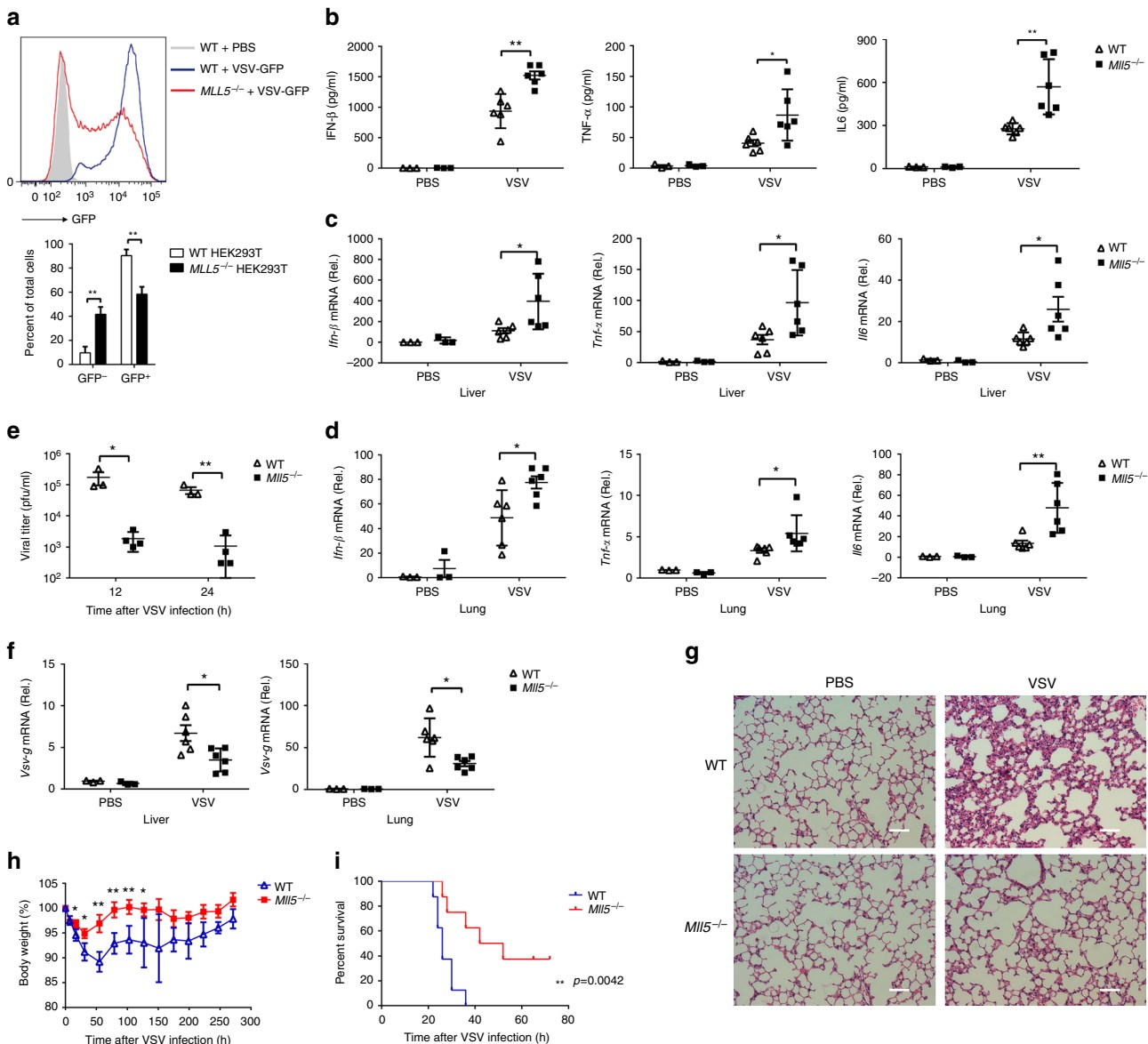

**Fig. 3** *MLL5* deficiency inhibits RNA virus replication both in vitro and in vivo. **a** Flow cytometry analysis of GFP fluorescence intensity in WT (blue) and *MLL5*<sup>−/−</sup> (red) HEK293T cells infected with VSV-GFP (MOI:0.1) for 20 h. WT HEK293T cells without infection served as a negative control (gray). Statistical results are shown at the bottom. **b–d** Six to nine weeks old WT or *Mll5*<sup>−/−</sup> littermates (n = 6) were intravenously infected with VSV-GFP (1 × 10<sup>7</sup> pfu/g). After 12 h, ELISA quantification of IFN-β, TNF-α and IL6 in sera (**b**). Expression of *Ifn-β*, *Tnf-α* and *Il6* mRNA after infection in the liver (**c**) and lung (**d**). *Gapdh* served as a control. **e** Six to nine weeks old WT (n = 3) or *Mll5*<sup>−/−</sup> littermates (n = 4) were intravenously infected with VSV-GFP (1 × 10<sup>7</sup> pfu/g), and sera were collected at 12 and 24 h after infection for plaque assay. **f** Expression of *Vsv-g* mRNA at 12 h after infection in the liver (left) and lung (right) treated as in **c** and **d**. *Gapdh* served as a control. **g** Hematoxylin–eosin staining of the lung sections from WT and *Mll5*<sup>−/−</sup> mice infected with VSV-GFP (1 × 10<sup>7</sup> pfu/g) for 12 h. Scale bar, 200 μm. **h** Six to nine weeks old WT or *Mll5*<sup>−/−</sup> littermates (n = 6) were intravenously infected with VSV-GFP (1 × 10<sup>6</sup> pfu/g) and monitored every 24 h after infection. **i** 6–9 weeks-old WT or *Mll5*<sup>−/−</sup> littermates (n = 8) were intravenously infected with VSV-GFP (2 × 10<sup>7</sup> pfu/g) and monitored every 6 h after infection. Data were representative of three independent experiments with similar results (**g**) or were from three independent experiments (**a–d**, **f**) or two independent experiments (**e**, **i**) or one experiment (**h**). Data were analyzed by Student's t-test (two-tailed) or log-rank (Mantel–Cox) test and were presented as mean ± SD (*p < 0.05,**p < 0.01)

higher in *Mll5*<sup>−/−</sup> than in wild-type MEFs after stimulation with VSV but not HSV-1. However, knockdown of RIG-I abrogated the augmented expression of *Ifn-β* and *Tnf-α* in *Mll5*<sup>−/−</sup> MEFs when stimulated with VSV (Fig. 4e). Therefore, these results suggested that MLL5-mediated suppression of innate immune responses against RNA viruses largely relies on RIG-I protein.

**MLL5 induces polyubiquitination and degradation of RIG-I.**
To test whether the transcription of *DDX58* gene was controlled by MLL5 protein, we detected the mRNA levels of *DDX58* in both wild-type and *MLL5*<sup>−/−</sup> HEK293T cells by qRT-PCR assay. The *DDX58* mRNA abundance remained unchanged in *MLL5*<sup>−/−</sup> HEK293T cells compared with wild-type HEK293T cells, suggesting that *MLL5*-deficiency-induced RIG-I accumulation occurs mainly at the post-translational level (Fig. 4f). To investigate whether the accumulation of RIG-I protein was due to inhibition of RIG-I degradation, we treated wild-type and *MLL5*<sup>−/−</sup> HEK293T cells with cycloheximide (CHX), an inhibitor of protein biosynthesis, and detected RIG-I protein levels for the

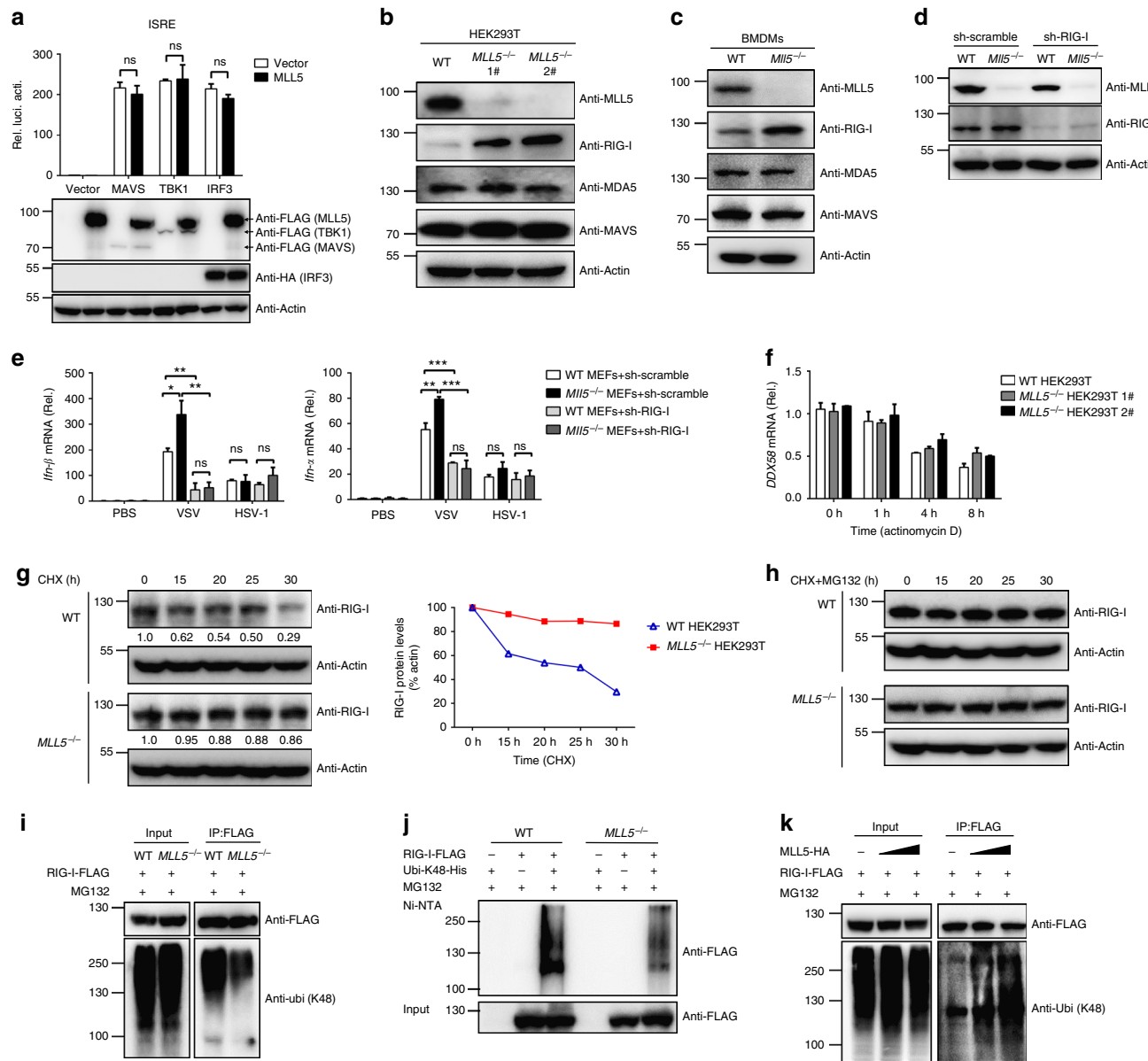

**Fig. 4** MLL5 deficiency results in accumulation of RIG-I protein. **a** Luciferase activity of ISRE promoter reporter in HEK293T cells transfected with expression plasmids for MAVS-FLAG (200 ng), TBK1-FLAG (200 ng) or IRF3-HA (200 ng) along with MLL5-FLAG (600 ng) or empty control vector (600 ng) for 24 h. Results were presented relative to the luciferase activity in control cells (transfected with luciferase reporter and empty vector). Immunoblot analysis of MAVS-FLAG, TBK1-FLAG, MLL5-FLAG or IRF3-HA are shown below. Actin served as a loading control. **b**, **c** Immunoblot analysis of MLL5, RIG-I, MDA5, and MAVS in WT and *MLL5*−/− HEK293T cells (**b**) or in BMDMs from WT and *Mll5*−/− mice (**c**). **d** Immunoblot analysis of MLL5 and RIG-I in WT and *Mll5*−/− MEFs with RIG-I knockdown. Actin served as a loading control. **e** Expression of *Ifn-β* and *Tnf-α* mRNA in WT and *Mll5*−/− MEFs with RIG-I knockdown infected with VSV-GFP (MOI:0.5) and HSV-1 (MOI:1) for 6 h. *Gapdh* served as a control. **f** Expression of *DDX58* mRNA in WT and *MLL5*−/− HEK293T cells treated with actinomycin D (1 μg/ml) for indicated times. *GAPDH* served as a control. **g**, **h** Immunoblot analysis of RIG-I in WT and *MLL5*−/− HEK293T cells treated with CHX (100 μg/ml) (**g**) or CHX (100 μg/ml) and MG132 (30 μM) (**h**). Actin served as a loading control. Relative band densities indicating protein levels are shown blown each band (**g**). Quantification of relative RIG-I protein levels is shown in the right panel. Band density indicating protein amount was quantified using Image J software. **i** Co-immunoprecipitation and immunoblot analysis of K48-linked polyubiquitination of RIG-I in WT and *MLL5*−/− HEK293T cells transfected with RIG-I-FLAG. **j** His-pull down and immunoblot analysis of K48-linked polyubiquitination of RIG-I in WT and *MLL5*−/− HEK293T cells cotransfected with RIG-I-FLAG and mutant ubiquitin K48-ubi-His. **k** Co-immunoprecipitation and immunoblot analysis of K48-linked polyubiquitination of RIG-I in HEK293T cells cotransfected with FLAG-RIG-I and MLL5-HA plasmids (0, 300, and 1000 ng). The HEK293T cells were treated with MG132 (5 μM) for 12 h before harvest (**i–k**). Data were representative of three independent experiments (**b–d**, **g–k**) or were from three independent experiments (**a**, **e**, **f**) and were analyzed by Student's *t*-test (two-tailed) and were presented as mean ± SD

indicated times. RIG-I protein underwent significant degradation upon CHX treatment in wild-type HEK293T cells. *MLL5* deficiency increased the half-life of endogenous RIG-I protein after CHX treatment (Fig. 4g and Supplementary Table 1). These results indicated that degradation of RIG-I protein was inhibited in *MLL5*$^{-/-}$ HEK293T cells. We next treated wild-type and *MLL5*$^{-/-}$ HEK293T cells with CHX and proteasome inhibitor MG132, and found that degradation of RIG-I protein was completely blocked in both cell types (Fig. 4h). These results suggest that MLL5 deficiency inhibits proteasome-mediated degradation of RIG-I.

Given that K48-linked polyubiquitination is required for proteasomal degradation of proteins, we determined whether the levels of K48-linked polyubiquitination of RIG-I were reduced

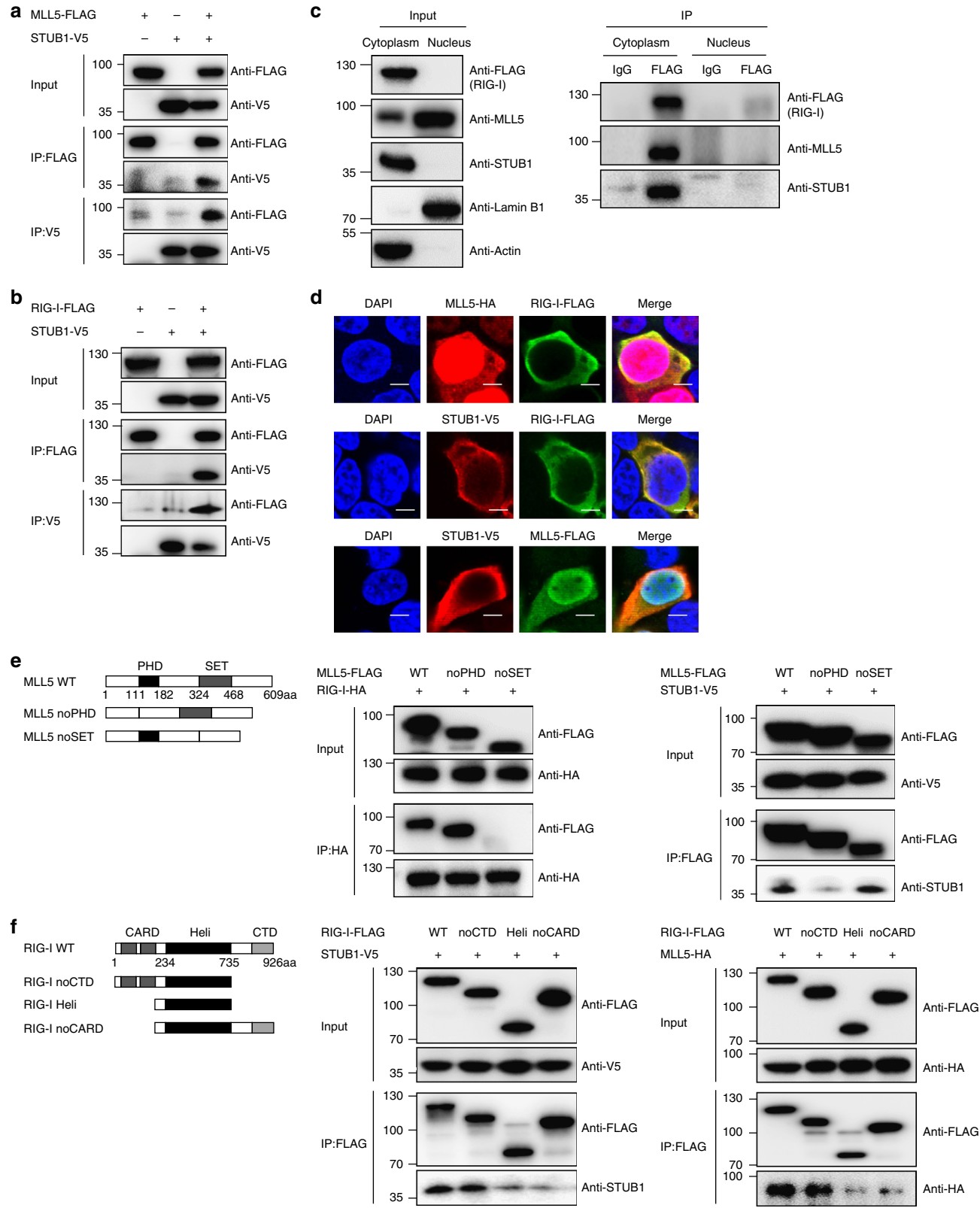

by *MLL5* deficiency. To test this, we transfected FLAG-tagged RIG-I vector into wild-type or *MLL5*$^{-/-}$ HEK293T cells. Forty-eight hours after transfection, FLAG-tagged RIG-I protein was precipitated with an anti-FLAG antibody, followed by immunoblot analysis with anti-K48-ubiquitin antibody. K48-linked polyubiquitination of RIG-I protein was significantly lower in *MLL5*$^{-/-}$ than in wild-type HEK293T cells (Fig. 4i). We cotransfected wild-type or *MLL5*$^{-/-}$ HEK293T cells with FLAG-tagged RIG-I vector and His-tagged mutants ubiquitin (K48) expression plasmids. Forty-eight hours after transfection, total proteins with K48-linked polyubiquitination were precipitated with Ni-NTA beads, followed by immunoblot analysis with an anti-FLAG antibody. Similarly, K48-linked polyubiquitination of RIG-I protein was markedly reduced in *MLL5*$^{-/-}$ HEK293T cells relative to that in wild-type HEK293T cells (Fig. 4j).

We next investigated whether overexpression of MLL5 had a substantial effect on K48-linked polyubiquitination of RIG-I. We cotransfected HEK293T cells with FLAG-tagged RIG-I and HA-tagged MLL5 plasmids. Forty-eight hours after transfection, FLAG-tagged RIG-I protein was precipitated with an anti-FLAG antibody, followed by immunoblot analysis with anti-K48-ubiquitin antibody. K48-linked polyubiquitination of RIG-I was increased by MLL5 in a dose-dependent manner (Fig. 4k). Together, these results demonstrate that MLL5 induces K48-linked polyubiquitination and proteasomal degradation of RIG-I.

**STUB1 associates with RIG-I and MLL5 in the cytoplasm**. The aforementioned findings raised the question of how MLL5 regulates K48-linked polyubiquitination and degradation of RIG-I. Because MLL5 itself lacks E3 ubiquitin ligase activity, we speculated that an E3 ligase may be involved in MLL5-mediated K48-linked polyubiquitination and proteasomal degradation of RIG-I. To test this, FLAG-tagged RIG-I protein was immunoprecipitated from HEK293T cells overexpressing FLAG-tagged RIG-I plasmids with anti-FLAG antibody, followed by mass spectrometry analysis to identify potential RIG-I-interacting proteins. MLL5 and several known E3 ligases for RIG-I were identified in the assay (Supplementary Table 2). Among the RIG-I-interacting proteins, we focused on STUB1 (also known as CHIP), which promotes RIG-I degradation (Supplementary Fig. 5a). We generated *STUB1*$^{-/-}$ HEK293T cells using a CRISPR-Cas9-based approach (Supplementary Fig. 5b), and examined the levels of endogenous RIG-I protein in *STUB1*$^{-/-}$ HEK293T cells. RIG-I protein was significantly increased in *STUB1*$^{-/-}$ HEK293T cells, similar to that observed in *MLL5*$^{-/-}$ HEK293T cells (Supplementary Fig. 5c). In contrast, K48-linked polyubiquitination of RIG-I protein was significantly decreased in *STUB1*$^{-/-}$ HEK293T cells compared with wild-type HEK293T cells (Supplementary Fig. 5d, e). The levels of STUB1 protein were not affected in *MLL5*$^{-/-}$ HEK293T cells (Supplementary Fig. 5c), suggesting that the STUB1 expression was not controlled by MLL5.

We next examined whether STUB1 interacted with RIG-I and MLL5. We cotransfected HEK293T cells with V5-tagged STUB1 vectors, FLAG-tagged RIG-I or FLAG-tagged MLL5 vectors, and

subjected them to immunoprecipitation and immunoblot analysis with anti-FLAG or anti-V5 antibodies. STUB1 could be co-immunoprecipated with both RIG-I and MLL5 (Fig. 5a, b).

RIG-I protein is located in the cytoplasm and detects cytosolic viral RNA during RNA virus infection. STUB1 is also located in the cytoplasm. In contrast, MLL5 protein is predominantly located in the nucleus. To understand the molecular mechanisms by which MLL5 induces K48-linked polyubiquitination and degradation of RIG-I, we separated the nuclear and cytoplasmic fractions of *DDX58* 3×FLAG knock-in HEK293T cells and detected the subcellular distributions of these proteins. RIG-I and STUB1 were retained in the cytoplasmic fraction and no protein was detected in the nucleus (Fig. 5c). Although most MLL5 protein was located in the nucleus, a small fraction was also detected in the cytoplasm (Fig. 5c). The subcellular localization of RIG-I, STUB1, and MLL5 was further supported by immunostaining and confocal microscopy assay in HEK293T cells (Fig. 5d). We then examined whether MLL5 interacted with RIG-I and STUB1 in the cytoplasm. Endogenous RIG-I protein was immunoprecipitated from the cytoplasm or nucleus of *DDX58* 3×FLAG knock-in HEK293T cells using anti-FLAG antibody, followed by immunoblotting with anti-MLL5 or anti-STUB1 antibodies, respectively. MLL5 and STUB1 proteins were co-immunoprecipitated with RIG-I protein only in the cytoplasm (Fig. 5c). These results indicated that cytoplasmic MLL5 was able to bind to RIG-I and STUB1.

We next sought to determine which domain was responsible for the binding among MLL5, STUB1, and RIG-I. Co-immunoprecipitation assays were performed using a variety of MLL5 and RIG-I truncation mutants (Fig. 5e, f). MLL5 protein has a single plant homeodomain (PHD) zinc finger and a Su(var) 3–9, Enhancer-of zeste, and Trithorax (SET) domain. We found that truncated MLL5 mutant lacking the SET domain abolished the capacity for RIG-I binding, indicating that the SET domain mediates interaction with RIG-I (Fig. 5e). By contrast, truncated MLL5 mutant lacking the PHD motif significantly weakened binding of MLL5 and STUB1, indicating that the PHD motif of MLL5 is required for binding to STUB1 (Fig. 5e). RIG-I protein contains multiple domains, including N-terminal CARD domains, a central DExD/H box helicase domain, and a CTD domain. Truncated RIG-I mutant lacking the CRAD domain impaired the capacity for MLL5 and STUB1 binding, indicating that the CARD domain of RIG-I is crucial for the interaction with MLL5 and STUB1 (Fig. 5f). Taken together, these results demonstrate that STUB1 associates with both RIG-I and MLL5 in the cytoplasm, and functions as a bona fide E3 ubiquitin ligase to induce K48-linked polyubiquitination and degradation of RIG-I.

**MLL5 is essential for interaction between RIG-I and STUB1**. The above results showed that RIG-I interacted with MLL5 and STUB1 in the cytoplasm, and deficiency in either MLL5 or STUB1 led to inhibition of K48-linked polyubiquitination and degradation of RIG-I. We determined whether MLL5 was required for interaction between RIG-I and STUB1. We examined the association between RIG-I and STUB1 in the presence or

**Fig. 5** STUB1 associates with RIG-I and MLL5 in the cytoplasm. **a** Co-immunoprecipitation and immunoblot analysis of HEK293T cell cotransfected with MLL5-FLAG and STUB1-V5. **b** Co-immunoprecipitation and immunoblot analysis of HEK293T cell co-transfected with RIG-I-FLAG and STUB1-V5. **c** Co-immunoprecipitation and immunoblot analysis of FLAG-tagged RIG-I, MLL5, and STUB1 in the nucleus and cytoplasm fractions in the *DDX58* 3×FLAG knock-in HEK293T cells. Actin served as a cytoplasmic control. Lamin B1 served as a nuclear protein control. **d** Confocal microscopy images of HEK293T cells cotransfected with RIG-I-FLAG, MLL5-HA, MLL5-FLAG or STUB1-V5. Scale bar, 5 μm. **e** Co-immunoprecipitation and immunoblot analysis of HEK293T cells cotransfected with RIG-I-HA or STUB1-V5 with MLL5-FLAG or MLL5-FLAG truncated vectors. **f** Co-immunoprecipitation and immunoblot analysis of HEK293T cells cotransfected with MLL5-HA or STUB1-V5 with RIG-I-FLAG or RIG-I-FLAG truncated vectors. Data are representative of three independent experiments with similar results

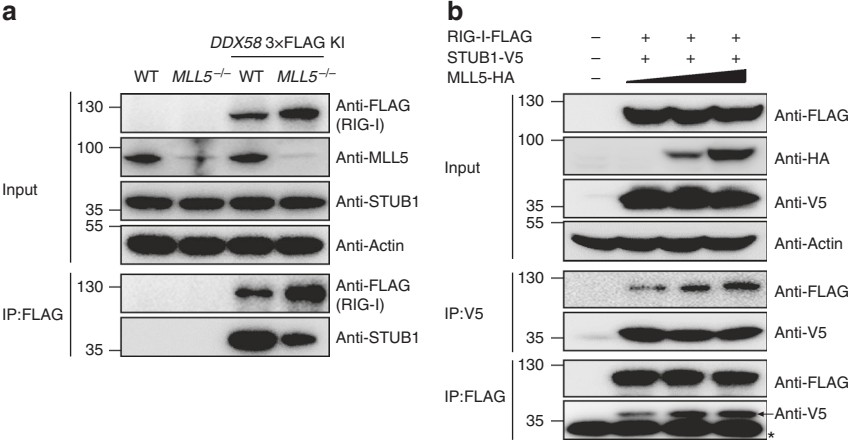

**Fig. 6** MLL5 facilitates interaction between RIG-I and STUB1. **a** Co-immunoprecipitation and immunoblot analysis of STUB1 in the WT and *MLL5*^−/−^ HEK293T cells and the *DDX58* 3×FLAG knock-in WT and *MLL5*^−/−^ HEK293T cells. Actin served as a loading control. KI, knock-in. **b** Co-immunoprecipitation and immunoblot analysis of HEK293T cell cotransfected with RIG-I-FLAG and STUB1-V5 with MLL5-HA plasmids (0, 1000, and 3000 ng). Arrow indicates STUB1 band. *IgG light chain. Data are representative of three independent experiments with similar results

absence of MLL5. FLAG-tagged RIG-I protein was precipitated from *DDX58* 3×FLAG knock-in wild-type HEK293T cells or *DDX58* 3×FLAG knock-in *MLL5*^−/−^ HEK293T cells using an anti-FLAG antibody, followed by western blot analysis with anti-STUB1 antibody. Co-immunoprecipitation between RIG-I and STUB1 was substantially attenuated in *MLL5*^−/−^ HEK293T cells compared with that in wild-type HEK293T cells (Fig. 6a). We next tested whether ectopic expression of MLL5 enhanced interaction between RIG-I and STUB1. We cotransfected HEK293T cells with FLAG-tagged RIG-I, V5-tagged STUB1 and HA-tagged MLL5 expression plasmids. Forty-eight hours after transfection, FLAG-tagged RIG-I protein or V5-tagged STUB1 protein was precipitated, followed by western blot analysis. RIG-I co-immunoprecipitation with STUB1 was markedly increased by ectopic expression of MLL5 protein in a dose-dependent manner (Fig. 6b). These results indicate that MLL5 plays an essential role in mediating the interaction between RIG-I and STUB1, thus ensuring STUB1-mediated K48-linked polyubiquitination and proteasomal degradation of RIG-I.

**Cytoplasmic MLL5 is critical for degradation of RIG-I.** We noted that the total MLL5 protein levels remained largely unchanged during VSV infection (Fig. 7a). However, the MLL5 levels in the cytoplasm were increased dramatically. Meanwhile, the MLL5 levels in the nucleus were reduced (Fig. 7a and Supplementary Table 3). Although CRM1 (also known as exportin 1) mediates the nucleocytoplasmic transportation of cargo proteins, we were unable to detect an association between CRM1 and MLL5, suggesting that the transport of nuclear MLL5 into the cytoplasm is mediated by a CRM1-independent mechanism.

To investigate whether nuclear MLL5 protein was involved in STUB1-mediated degradation of RIG-I through a transcription-dependent mechanism in the nucleus, we generated MLL5 combined with two nuclear localization signals (NLSs) to ensure its full nuclear compartmentalization (Fig. 7b). MLL5-NLS proteins were only detected in the nucleus, whereas wild-type MLL5 proteins were detected in both the nucleus and cytoplasm (Fig. 7c). The nuclear distribution of MLL5-NLS protein was confirmed by immunostaining and confocal microscopy assay in HEK293T cells (Fig. 7d). These results show that two NLS signals are efficient at targeting MLL5-NLS protein in the nucleus. We next transfected *MLL5*^−/−^ HEK293T cells with HA-tagged wild-type MLL5 or MLL5-NLS vectors, and detected the levels of

endogenous RIG-I protein. The levels of endogenous RIG-I proteins were markedly reduced after transfection of wild-type MLL5 but not MLL5-NLS protein (Fig. 7e and Supplementary Table 4). We cotransfected *MLL5*^−/−^ HEK293T cells with wild-type MLL5 or MLL5-NLS vectors along with STUB1 vectors, and observed a decrease in endogenous RIG-I protein in *MLL5*^−/−^ HEK293T cells transfected with wild-type MLL5 and STUB1 in a dose-dependent manner (Fig. 7f). By contrast, the levels of endogenous RIG-I protein remained unchanged in *MLL5*^−/−^ HEK293T cells transfected with MLL5-NLS and STUB1 (Fig. 7f). Thus, these results demonstrate that cytoplasmic but not nuclear MLL5 is required for STUB1-mediated degradation of RIG-I.

**Discussion**

In this study, we show that the Trithorax group protein MLL5 acts as a specific negative regulator for RIG-I-triggered innate antiviral immune responses by promoting STUB1-mediated K48-linked polyubiquitination and proteasomal degradation of RIG-I in the cytoplasm. MLL5 deficiency leads to accumulation of RIG-I protein in human cells and in mice in vivo. As a consequence, *Mll5*-deficient mice produced more type I IFNs and proinflammatory cytokines in response to RNA virus infection and exhibited enhanced innate immune responses, and reduced viral load and morbidity in vivo. These findings reveal a previously unrecognized role of MLL5 in regulating antiviral innate immune responses.

MLL5 belongs to an evolutionarily conserved Trithorax family. Trithorax family members such as MLL1–4 positively regulate transcription of target genes through deposition of activating H3K4 methylation marks at the promoters[22]. Whether MLL5 exerts intrinsic H3K4 methyltransferase activity is still under debate, however, previous studies have suggested that MLL5 contributes to the transcriptional regulation of several host or viral genes, mainly in the nucleus[14]. In this study, we showed that a fraction of MLL5 localized in the cytoplasm plays an important role in antiviral innate immunity. Cytoplasmic MLL5 interacts specifically with the cytosolic RNA receptor RIG-I and its E3 ubiquitin ligase STUB1. RIG-I–STUB1 interaction was attenuated in the absence of MLL5, which suggests that cytoplasmic MLL5 provides an adaptor for the association between RIG-I and STUB1. Consequently, MLL5 deficiency inhibited the STUB1-mediated degradation of RIG-I, and the levels of RIG-I were significantly elevated. These results might explain why *Mll5*-deficient mice produced more type I IFNs and exhibited

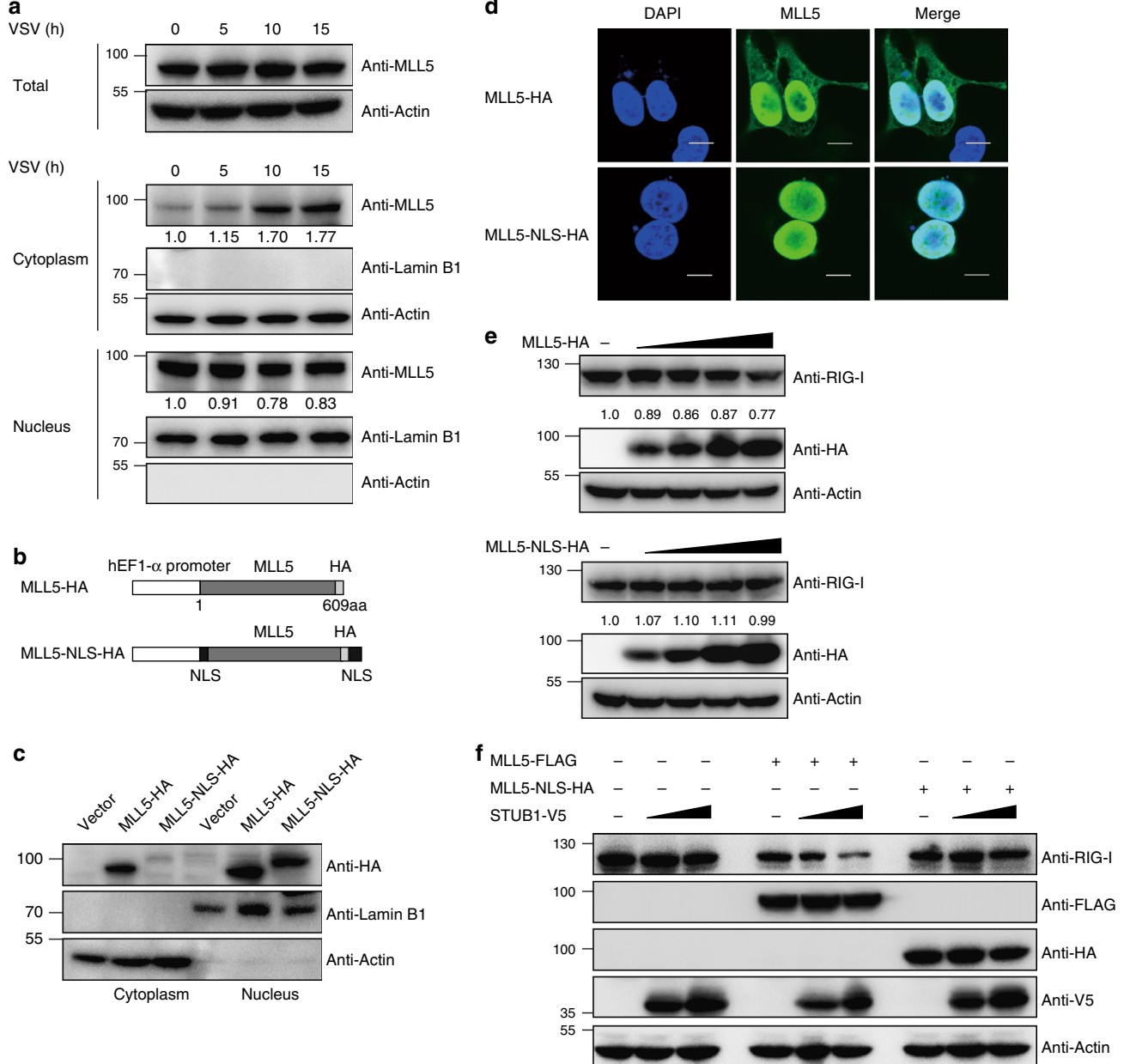

**Fig. 7** Cytoplasmic MLL5 is crucial for degradation of RIG-I. **a** Immunoblot analysis of MLL5 in total or in the nuclear and cytoplasmic fractions in WT HEK293T cells infected with VSV-GFP (MOI:1). Actin served as a cytoplasmic control. Lamin B1 served as a nuclear protein control. Relative band densities indicating protein levels are shown blown each band. Band density indicating protein amount was quantified using Image J software. **b** Schematic representation of MLL5-NLS-HA with NLS signals at N and C terminals. **c** Immunoblot analysis of MLL5 in the nuclear and cytoplasmic fractions in HEK293T cells transfected with MLL5-HA or MLL5-NLS-HA vectors. Actin served as a cytoplasmic control. Lamin B1 served as a nuclear protein control. **d** Confocal microscopy images of HEK293T cells transfected with MLL5-HA or MLL5-NLS-HA plasmids. Scale bar, 10 μm. **e** Immunoblot analysis of RIG-I in *MLL5*$^{-/-}$ HEK293T cells transfected with different amount of MLL5-HA or MLL5-NLS-HA plasmids (0, 300, 600, 1000, and 2000 ng). Actin served as a loading control. Relative band densities indicating protein levels are shown below each band. Band density indicating protein amount was quantified using Image J software. **f** Immunoblot analysis of RIG-I in *MLL5*$^{-/-}$ HEK293T cells transfected with empty vector, MLL5-FLAG plasmids or MLL5-NLS-HA plasmids, with STUB1-V5 plasmids (0, 400, and 1000 ng). Actin served as a loading control. Data are representative of three independent experiments with similar results

enhanced antiviral innate immune responses upon RNA virus infection.

DNA damage induces type I IFN through the cytosolic DNA sensor STING to promote innate antimicrobial immunity[23–26]. A recent study has demonstrated that *Mll5*-deficiency causes accumulation of DNA damage and ROS in HSPCs, and the DNA damage can trigger IFN-1 response and lead to proliferation and malfunction of HSPCs by the IFN-1/Bid/ROS pathway[18]. The

levels of type I IFNs in the sera and bone marrow supernatants were not significantly increased in their *Mll5*-deficient mice[18]. Similarly, no significant increase in the IFN-β and TNF-α levels in the sera was detected in our *Mll5*-deficient mice (Supplementary Fig. 6). Moreover, IFN-β and TNF-α induction by stimulation with DNA virus, as well as that of other PAMP ligands was not affected by *Mll5* deficiency (Fig. 1a–c). Therefore, accumulation of RIG-I protein in *Mll5*-deficient cells and the resistance of *Mll5*-

deficient mice to VSV infection were unlikely due to DNA-damage-induced type I IFN responses. Moreover, knockdown of RIG-I abrogated the augmented expression of IFN-β and TNF-α and antiviral activity induced by stimulation with RNA virus in *Mll5*-deficient MEFs. These results further indicated that, although RIG-I-independent mechanism could not be fully excluded, *Mll5*-deficiency-induced augment of expression of IFN-β is largely dependent on RIG-I (Fig. 4d, e). Nevertheless, our current study reveals a new mechanism of MLL5 in controlling antiviral innate immune responses upon RNA virus infection.

In this study, we ectopically expressed MLL5 protein carrying two NLS sequences in *MLL5*^−/− HEK293T cells, so that all MLL5 protein was forcibly localized to the nucleus. The absence of cytoplasmic MLL5 inhibits STUB1-mediated RIG-I degradation, indicating that cytoplasmic, but not nuclear MLL5 is involved in antiviral innate immunity. We observed an obvious increase of cytoplasmic MLL5 during VSV infection. These results indicate that nuclear MLL5 may translocate to the cytoplasm upon RNA virus infection. A recent study has demonstrated that FBXW7, an E3 ubiquitin ligase, translocates from the nucleus to the cytoplasm to stabilize RIG-I upon virus infection[27]. FBXW7 contains a nuclear export signal (NES), which could interact with CRM1 to mediate nucleocytoplasmic transport of FBXW7. However, MLL5 protein lacks an NES signal, suggesting that a yet-unknown NES-independent mechanism mediates the nucleocytoplasmic translocation of MLL5 upon RNA virus infection. Nevertheless, these data provided compelling evidence that MLL5 protein translocates from the nucleus to the cytoplasm to induce STUB1-mediated degradation of RIG-I during VSV infection, thereby restricting RIG-I-triggered innate immune responses. Thus, the nucleocytoplasmic translocation of nuclear proteins, such as FBXW7 and MLL5, illustrates an important mechanism for precise regulation of innate immune responses during virus infection.

Epigenetic modifying factors are increasingly shown to be involved in modulation of immune and inflammation responses via control of gene expression of immune signaling molecules[28]. For example, EZH1, a component for polycomb repressive complex (PRC)2, promotes TLR-triggered inflammatory cytokine production by suppressing the negative regulator Toll interacting protein (TOLLIP) by inducing repressive H3K27 modification at the TOLLIP promoter[29]. In addition, ASH1L, a histone methyltransferase, inhibits LPS-induced production of proinflammatory cytokines by directly enhancing expression of the ubiquitin-editing enzyme A20, by inducing H3K4 modification at the A20 promoter[30]. Our study provides new evidence that, upon viral infection, epigenetic regulator could translocate from the nucleus to the cytoplasm to target directly the signaling molecules of the innate immune responses (Supplementary Fig. 7). To the best of our knowledge, MLL5 is the first epigenetic factor to be identified as being directly involved in innate immune responses in the cytoplasm. Further investigations are expected to determine whether any other epigenetic regulators exert a similar role in regulating innate immune responses outside the nucleus.

In summary, we show a previously unrecognized role of MLL5 in suppressing RLR-mediated innate immune responses by promoting RIG-I degradation through E3 ubiquitin ligase STUB1 in the cytoplasm. Our findings suggest a new potential therapeutic target for modulating innate immune responses to limit RNA virus infection or treat excessive autoimmune diseases.

## Methods

**Reagents.** 5′-pppRNA(tlrl-3prna) was purchased from InvivoGen and poly(I:C) (P1503) was purchased from Sigma. 5′-pppRNA and poly(I:C) were used at a final concentration of 0.2 and 1 μg/ml, respectively. MG132 was purchased from Selleckchem (S2619). Cycloheximide(CHX) was purchased from Sigma (C7698).

MG132 and CHX were used at a final concentration of 30 μM and 100 μg/ml, respectively. The antibody specific for MLL5 (AP14173a, Rabbit, 1:500) was purchased from Abgent. The antibodies specific for FLAG M2 (F1804, Mouse, 1:5000) and Actin (A5316, Mouse, 1:5000) were purchased from Sigma. The antibody specific for HA (11583816001, Mouse, 1:3000) was purchased from Roche. The antibodies specific for RIG-I(D14G6)(3743S, Rabbit, 1:500), P65 (8242S, Rabbit, 1:1000), p-P65 (Ser 536)(93H1, Rabbit, 1:1000), and K48-ubi (4289, Rabbit, 1:1000) were purchased from Cell Signaling Technology. The antibodies specific for V5 (SC-271944, Mouse, 1:5000), His (SC-8036, Mouse, 1:2000) and Lamin B1 (SC-20682, Rabbit, 1:1000) were purchased from Santa Cruz Biotechnology. The antibody specific for p-IRF3 (Ser 396)(ab138449, Rabbit, 1:1000) was purchased from Abcam. The antibodies specific for MDA5 (21775-1-AP, Rabbit, 1:2000), MAVS (14341-1-AP, Rabbit, 1:5000) and IRF3(11312-1-AP, Rabbit, 1:5000) were purchased from Proteintech. The antibody specific for STUB1 (D154048, Rabbit, 1:1000) was purchased from BBI. The anti-mouse Alexa Fluor 555(A21422, 1:750) and anti-rabbit Alexa Fluro 488 (A110081, 1:500) were purchased from Invitrogene.

**Plasmid.** Plasmids expressing MLL5 (tagged with 3×Flag or HA) in mammalian cells were constructed by in-frame insertion of a human MLL5 cDNA into pcDEF3 expression vector. Plasmids expressing STUB1 (tagged with V5) in mammalian cells were constructed by in-frame insertion of a human STUB1 cDNA into pcDNA3.1 expression vector. The RIG-I-FLAG, ISRE firefly luciferase reporter and *Renilla* luciferase reporter plasmids were kindly provided by Jin Zhong (Institut Pasteur of Shanghai). The MAVS-FLAG plasmid was kindly provided by Guangxun Meng (Institut Pasteur of Shanghai). The TBK1-FLAG and IRF3-HA plasmids were kindly provided by Hui Xiao (Institut Pasteur of Shanghai). The Ubi-K48-His plasmid was kindly provided by Bin Li (Shanghai Institute of Immunology, Shanghai Jiaotong University). The IFN-β and NF-κB firefly luciferase reporter plasmids were kindly provided by Chen Wang (China Pharmaceutical University, Nanjing). The NLS sequence in MLL5-NLS-HA plasmid were 5′-ccaaagaa-gaagcggaaggtcggt-3′ and 5′-aaaaggccggcggccacgaaaaaggccggccaggcaaaaaagaaaaag-3′.

**Mice.** The *Mll5*-deficient mice were produced by microinjecting Cas9 mRNAs (50 ng/μl) and gRNA (20 ng/μl) into fertilized eggs from C57BL/6J mice. The genotyping of the *Mll5*-deficient mice was confirmed by sequencing of the PCR fragments in the gRNA-targeting region amplified from genomic DNA isolated from tail tips using the following primers: forward 5′-gactttcatctactagacag-3′ and reverse 5′-agttgcatctgatagaggca-3′. Genotyping of WT and *Mll5*−/− mice was performed with the following primers: forward primer 5′-caggtgcatatgtggttttt-3′ and reverse 5′-aggattacatctaaagatct-3′ for WT mice; forward primer 5′-accaggtggttttaca-3′ and reverse primer 5′- aggattacatctaaagatct -3′ for for *Mll5*−/− mice. The newly constructed *Mll5*−/− mice were maintained as heterozygotes by backcrossing with WT C57BL/6J mice for at least four generations. C57BL/6J mice were purchased from Jax lab. All of the mice were bred and maintained under specific-pathogen-free animal facility in Institut Pasteur of Shanghai.WT and *Mll5*−/− mice used in this study were littermates with same sex on C57BL/6J background at ages indicated in specific experiments. All procedures were conducted in compliance with the guidelines from the Institutional Animal Care and Use Committee and were approved by the Institutional Ethics Committee in Institut Pasteur of Shanghai, Chinese Academy of Sciences.

**sgRNA design and in vitro transcription.** The vector expressing CAS9 and sgRNA was purchased from addgene (#42230) and was digested with BbsI, then the linearized vector was gel purified. sgRNAs were designed using the CRISPR tool (http://crispr.mit.edu/). A pair of oligos for each site was annealed and ligated to the linearized vector. T7 promoter was added to Cas9 coding reagion and sgRNAs template by PCR amplication. The T7-Cas9 plasmid was digested with Sal I and gel purified, then used as the template for in vitro transcription using mMESSAGE mMACHINE T7 ULTRA kit (Life Technologies, #AM1345). The T7-sgRNA plasmid was digested with BamH I and gel purified, then used as the template for in vitro transcription using MEGAshortscript T7 kit (Life Technologies, #AM1354). Both the Cas9 mRNA and the sgRNAs were purified using MEGAclear kit (Life Technologies, #AM1908) and eluted in Rnase-free water.Oligonucleotides used in this study are listed as follows: *MLL5*-1#-F: caccgctgctggtatataactggt and *MLL5*-1#-R: aaacaccagttatataccagcagc; *Mll5*-F or *MLL5*-2#-F: caccgatgtaaccaggtg-catatg and *Mll5*-R or *MLL5*-2#-R: aaaccatatgcacctggttcacat; *STUB1*-1#-F: caccgctggacgggcagtctgtga and *STUB1*-1#-R: aaactcacagactgcccgtccagc; *STUB1*-2#-F: caccggggcagtgccagctggaga and *STUB1*-2#-R: aaactctccagctggcactgcccc; *DDX58*-1#-F: caccgattgaggacctgatatcatt and *DDX58*-1#-R: aaacaatgatatcaggtcctcaatc.

**BM transplantation assay.** For BM transplantation assay, $8 \times 10^6$ BM cells from WT or *Mll5*−/− littermate mice (5–6 weeks old) were injected into the tail veins of 7–8 weeks old female C57BL/6J mice that had been lethally irradiated (9.5 Gy, X-ray) 24 h previously.

**Cells and viruses.** All cell cultures were maintained in a humidified atmosphere at 37 °C with 5% $CO_2$. The HEK293T cells (ATCC, CRL-11268) were cultured in DMEM medium (Dulbecco's modified Eagle's medium) supplemented with 10%

fetal bovine serum (FBS), non-essential amino acids, and β-mercaptoethanol. Sendai virus (SeV) was kindly provided by Jin Zhong (Institut Pasteur of Shanghai). Vesicular stomatitis virus (VSV) (Indiana strain) and herpes simplex virus 1 (HSV-1) (KOS strain) were provided by Hui Xiao (Institut Pasteur of Shanghai).

**Isolation of mouse embryonic fibroblasts and macrophages.** Mouse embryonic fibroblasts from wild-type and *Mll5*-deficient mice were prepared from E13.5 day embryos and cultured in DMEM medium supplemented with 10% FBS, non-essential amino acids and β-mercaptoethanol. Peritoneal macrophages were harvested from wild-type and *Mll5*-deficient mice 3 days after thioglycollate (3.8%, Sigma) injection and cultured in RPMI medium 1640 supplemented with 10% FBS and L-glutamine. BMDMs were isolated from the femur of wild-type and *Mll5*-deficient mice and cultured in 10-cm Petri dish at 37 °C for 7 days. At days 3 and 5, 5 ml RPMI medium 1640 (supplemented with 10% FBS, L-glutamine, and 30% L929 supernatant) was added.

**Enzyme-linked immunosorbent assay.** The concentrations of cytokines in cell culture supernatants and sera from VSV infected mice were measured using mouse IFN-β (439407, BioLegend) and IL-6 (431301, BioLegend), mouse TNF-α (88-7324-88, eBioscience) enzyme-linked immunosorbent assay (ELISA) Kits according to the manufacturer's instructions.

**RNA quantification.** Total RNA was extracted with TRIzol reagent following the manufacturer's instructions (Invitrogen). Reverse-transcription products were amplified by an ABI 7900HT Fast Real-Time PCR System (Applied Biosystems) using the SYBR Green PCR Master Mix (DRR420A, Takara) and data were normalized to the expression of the *Gapdh* in each individual sample. The $2^{-\Delta\Delta Ct}$ method was used to calculate relative expression changes. Specific primers used for RT–PCR assays are listed as follows: human *DDX58*-forward: 5′-ggacgtggcaaaa-caaatcag-3′ and *DDX58*-reverse: 5′-gcaatgtcaatgccttcatca-3′; human *IFN-β*-forward: 5′-ccaacaagtgtctcctccaa-3′ and *IFN-β*-reverse: 5′-atagtctcattccagccagt-3′; human *TNFα*-forward: 5′-cgagtgacagacctgtagc-3′ and *TNFα*-reverse: 5′-ggtgtgggtgaggagcacat-3′; human *GAPDH*-forward:5′-catgttcgtcatgggtgtgaacca-3′ and *GAPDH*-reverse: 5′-agtgatggcatggactgtggtcat-3′; mouse *Ifn-β*-forward: 5′-ctgcgttcctgctgtgcttctcca-3′ and *Ifn-β*-reverse: 5′-ttctccgtcatctcctatgggatc-3′; mouse *Tnfα*-forward: 5′-gtccccaaagggatgagaagtt-3′ and *Tnfα*-reverse: 5′-gtttgctac-gacgtgggctaca-3′; mouse *Mx1*-forward: 5′-ctgagatgacccagcacctgaa-3′ and *Mx1*-reverse: 5′-ctccaggaaccagctgcacttac-3′; mouse *Il6*-forward: 5′-agataagctggagtcaca-gaaggag-3′ and *Il6*-reverse: 5′-cgcactaggtttgccgagtag-3′; mouse *Gapdh*-forward: 5′-tggagaaacctgccaagtatga-3′ and *Gapdh*-reverse: 5′-ctgttgaagtcgcaggagacaa-3′; VSV-G-forward: 5′-caagtcaaatgcccaagagtcaca-3′ and VSV-G-reverse: 5′-tttccttgcattgttctacagatgg-3′.

**Immunoblot analysis and co-immunoprecipitation.** For immunoblot analysis, cells or tissues were lysed with lysis buffer (1% NP-40, 20 mM HEPES, 20 mM KCl, 150 mM NaCl, 5 mM EDTA, 1 mM Na$_3$VO$_4$ and complete protease inhibitor cocktail (Roche), pH 7.4) on ice for 30 min. For co-immunoprecipitation (co-IP), whole-cell extracts were collected 48 h after transfection and lysed in lysis buffer on ice for 30 min. After centrifugation for 10 min at 13,000×*g*, 4 °C, supernatants were collected and incubated with Protein-G Sepharose beads coupled to specific antibodies for 2 h or overnight with rotation at 4 °C, beads were washed 3× times with lysis buffer. Bound proteins were eluted by boiling 10 min with sample buffer (50 mM Tris-HCl (pH 6.8), 2% SDS, 10% glycerol, 0.1% bromophenol blue and 1% β-mercaptoethanol). For immunoblot analysis, immunoprecipitates or whole-cell lysates were separated by SDS–PAGE, electro-transferred to PVDF membranes and blocked for 1 h with 5% no-fat milk solution, followed by blotting with the appropriate antibodies and detection by enhanced chemiluminiscence (ECL). Uncropped scans of western blots are provided in Supplementary Figs. 8 and 9.

**Mass spectrometry.** HEK293T cells were transfected with RIG-I-FLAG plasmid and pcDNA3.1 vector for 48 h. The cell lysates were immunoprecipitated by Sepharose beads covalently conjugated to FLAG specific mAb (A2220, Sigma). Bound proteins were competitively eluted by 3×FLAG peptide (F4799, Sigma) and used as the sample for mass spectrometry.

**Ubiquitination assay and His pull-down assay.** For analysis of the ubiquitination of RIG-I in HEK293T cells, HEK293T cells were transfected with plasmids expressing RIG-I-FLAG and were treated with 5 μM MG132 for 12 h before harvesting, then whole-cell extracts were immunoprecipitated with the FLAG-specific antibody and analyzed by immunoblot with anti-K48-ubi. HEK293T cells for Ubiquitin-His pull down were transfected with plasmids expressing RIG-I-FLAG and K48-ubiquitination-His and treated with 5 μM MG132 for 12 h before harvesting, then lysed in pH 8.0 urea lysis buffer (8 M Urea, 100 mM Na$_2$HPO$_4$, 10 mM Tris-HCl, pH8.0, 0.2% TritonX-100, 10 mM Imidazole) for 20 min. Spin at 12000 rpm, 30 min. Cell lysates were incubated with Ni-NTA beads (Qiagen) for 4 h at room temperature. The beads were sequentially washed twice with pH 8.0 urea buffer, pH 6.3 urea buffer (8 M Urea, 100 mM Na$_2$HPO,10 mM Tris-HCl, pH 6.3, 0.2% TritonX-100, 10 mM Imidazole) and pH 8.0 wash buffer (10 mM Tris-HCl, pH 8.0, 100 mM NaCl, 20% Glycerol, 1 mM DTT, 10 mM Imidazole). At last, preloading buffer (62.5 mM Tris-HCl, pH 6.8, 15%SDS, 8 M Urea, 10% glycerol,

100 mM DTT) and 5× loading buffer was added and incubated for 30 min at 30 °C. The samples were separated on SDS-PAGE gel.

**RNA interference.** Specific knockdown of RIG-I in MEF cells was induced by expression of corresponding shRNAs (shRNAs were cloned into the lentivirus vector pLKO.1 plasmid). pLKO.1 plasmids were used together with packaging plasmids (pSPAX2 and pMD2.G) to cotransfect HEK293T cells cultured in 10 cm dishes. Lentiviral stocks were used to infect the MEF cells. In these experiments, MEFs were infected with lentivirus encoding target sequence against RIG-I (5′-GTCAGAATCTCAGTCAGAATC-3′) and scramble sequence (5′-TTCTCCGAACGTGTCACGTAC-3′), followed by cell selection through puromycin (1 μg/ml).

**Luciferase assays.** HEK293T ($1 \times 10^5$ cells/well) in 48-well plates were transfected with MLL5 (1 μg) or controls plasmid (1 μg) together with IFN-β or NF-κB or ISRE-firefly luciferase reporter (200 ng) and *Renilla* luciferase reporter (5 ng). Twenty-four hours later, the cells were stimulated with poly(I:C) (1 μg/ml) for 16 h, and luciferase activities were measured with the Dual Luciferase reporter gene assay kit (RG-027, Beyotime). Data were normalized by calculating the ratio between firefly luciferase activity and *Renilla* luciferase activity.

**Immunofluorescence confocal microscopy.** HEK293T cells grown on glass coverslips were fixed for 15 min using 4% paraformaldehyde in PBS, permeabilized in PBS containing 0.25% Triton X-100 for 10 min and blocked with PBST solution containing 1% BSA, then the cells were stained with the indicated primary antibodies followed by incubation with fluorescent-dye-conjugated secondary antibodies. Nuclei were stained with DAPI (Sigma). Images were acquired using Olympus FV-1200.

**Viral infection in vitro.** Mouse macrophages, MEFs or HEK293T cells were plated 24 h before infection. Cells were infected with VSV (1 MOI), HSV-1 (5 MOI), or SeV (1 MOI) for 2 h in medium without FBS, after infection, cells were washed with PBS and then medium was added with FBS. The cells and cell-free supernatants were harvested at the indicated times. Total cellular RNA was extracted and VSV RNA replicates were examined by qRT-PCR as described. The supernatants was stored at −80 °C until ELISA assay. Each virus infection was performed in triplicate.

**Viral infection in vivo and plaque assay.** For in vivo viral infection, 6–8 weeks old and sex-matched wild-type and *Mll5*-deficient littermate mice were infected with high dose ($1 \times 10^7$ pfu/g) or low-dose ($1 \times 10^6$ pfu/g) VSV via intravenous injection into the tail vein. Cytokines production in the sera was measured by ELISA. The virus titers in the sera were determined by standard plaque assays. The cytokines expression and viral RNA replicates in lung and liver were examined by Quantitative RT-PCR as described. For the survival experiments, mice were monitored for survival after VSV infection. Lungs from control or virus infected mice were dissected, fixed in 10% phosphate-buffered formalin, embedded into paraffin, sectioned, stained with hematoxylin–eosin solution and examined by light microscopy for histological changes.

For plaque assay, sera samples obtained after VSV infection were serially diluted and then added into confluent Vero cells cultured in 24-well plates. At 2 h after infection, the supernatant was removed and cells was washed with PBS twice. The cells was overlaid by 3% methylcellulose. Four day later, the overlay was removed and cells were fixed with 4% formaldehyde for 1 h and stained with 0.5% crystal violet. The plaque was counted, averaged, and viral titers were calculated accordingly.

**Statistical analysis.** Statistical significance between groups was determined by two-tailed student's *t*-test. Differences were considered to be significant when $p < 0.05$ (*$p < 0.05$, **$p < 0.01$, ***$p < 0.001$, ****$p < 0.0001$). For mice survival studies, Kaplan–Meier survival curves were generated and analyzed for statistical significance with GraphPad Prism 5.0.

**Data availability.** All relevant data are available from the authors upon request.

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

## Acknowledgements

We thank Drs. Chen Wang, Hong Tang, Ronggui Hu, Ying Wan for helpful discussion. We thank Drs. Catherine C.L. Wong and Chao Peng from mass spectrometry system at the National Facility for Protein Science Shanghai (NFPSS) and National Center for Protein Science Shanghai (NCPSS) for MS analysis and data processing. This work was supported by the National Basic Research Program of China (2015CB964900), the National Natural Science Foundation of China (31670906, 31471207, 81270618), and the Strategic Priority Research Program of Chinese Academy of Sciences (XDPB03).

## Author contributions

P.Z. and Y.Z. designed the experiments and analyzed the data; W.Z., X.D., L.L., X.W., X.Y., and X.H. performed the experiments; G.M., H.X., B.L., J.Z., F.H., and L.D. contributed new reagents/analytical tools. P.Z. and Y.Z. wrote the paper.

## Additional information

**Competing interests:** The authors declare no competing interests.

