## [Peer Review File(PDF 306 kb) · Nature Communications]

Reviewers' comments:

Reviewer #1 (Remarks to the Author):

In their study, Zhong et al. provide an exciting set of novel data establishing a surprising connection between the deficiency of MLL5 - a presumed epigenetic regulator - and an enhanced anti-viral immune response. In particular, the authors demonstrate that Mll5-deficient mice newly generated for the study by CRISPR/Cas technology (3 independent lines of Mll5-deficient mice have been described previously by others – references 15 -17 in the manuscript) and cell lines derived thereof exhibit elevated levels of IFNbeta, TNFalpha and IL6 cytokines when infected with RNA, but not DNA viruses. Subsequent in vivo studies reveal that Mll5-deficient mice are markedly more resistant to vesicular stomatitis virus (VSV) infection when compared with wildtype controls. In an extended series of well-controlled biochemical experiments with two primary cell types (MEFs, BMDMs) derived from Mll5 knock-out mice and human HEK293T cells (made Mll5-deficient via CRISPR/Cas), the authors provide conclusive evidence that cytoplasmic MLL5 protein destabilizes the intracellular pattern recognition receptor RIG-I by recruiting the deubiquitinase STUB1, which boosts K48 ubiquitination and proteasomal degradation of RIG-I. The authors propose that the observed stabilization of RIG-I in the absence of Mll5 provides the key mechanism to explain the increased VSV resistance of Mll5^{-/-} mice, as documented here for the first time. The topic investigated and the results obtained are of high scientific interest and should stimulate researchers from distinct fields of investigation. The experiments seem very well performed and the results are convincing. If the authors can address the few concerns mentioned below, I strongly recommend publication.

Comments/Concerns – Major:

1. INSUFFICIENT CHARACTERIZATION OF NEWLY GENERATED MLL5 MOUSE MUTANTS: Mll5-mutant mice were newly generated for this study by deleting 8 nucleotides in exon 3 using CRISPR/Cas9 technology. The authors state in the “Methods” section: The genotype “of the Mll5-deficient mice was confirmed by sequencing of the PCR fragments in the gRNA-targeting region amplified from genomic DNA isolated from tail tips using the following primers: ...”. No experimental data documenting the effect of this rather limited mutation on MLL5 protein expression are provided. The mouse Mll5 gene is composed of at least 25 coding exons giving rise to a full-length protein of 1868 amino acids. Without data on protein expression, it cannot be excluded that disruption of exon 3 is partially neutralized by in-frame splicing from exon 2 to exon 6, 7 or 8, which would give rise to a somewhat truncated MLL5 protein of significant length lacking the PHD domain, but still harboring the SET domain. The authors should use antisera directed against the amino- and/or carboxy-terminus of mouse MLL5 to quantify the effect of the CRISPR/Cas9-induced exon 3 mutation on overall protein expression. Antibodies directed against epitopes encoded by exons 3, 4 or 5, like the antiserum (AP14173) used in this study, are clearly not appropriate to document absence of truncated MLL5 protein variants.

Moreover, the authors should provide data documenting to what extent the phenotype of their newly generated Mll5-mutant mice corresponds to the consensus phenotype of the three separate Mll5-deficient mouse strains published previously (references 15 – 17 in the manuscript) - phenotypic features like mild growth retardation, male sterility, strongly reduced thymocyte and splenocyte numbers, radiation sensitivity etc. Without a more comprehensive characterization of the newly generated Mll5-mutant strain the possibility remains that exon 3 disruption has produced either a hypomorph or a truncated protein with dominant negative activity. Similar arguments could be made for the MLL5-mutated HEK293T cell line used for most biochemical studies.

The authors should also state how many times the original exon 3 mutation has been back-crossed onto C57BL/6 to minimize potential effects of off-target mutations on the observed phenotype. In case first-generation mouse mutants were used for experiments, the potential problem with off-target mutations in this study should be briefly debated in the “Discussion” section.

2. THE DATA DO NOT FORMALLY ESTABLISH A CAUSAL RELATIONSHIP BETWEEN ELEVATED RIG-I PROTEIN LEVELS AND THE IMPROVED DEFENSE OF MLL5-DEFICIENT MICE AND CELLS AGAINST RNA VIRUSES: In line with the overall “tone” of the manuscript, the authors state in the “Discussion” (page 18, line 352) “MLL5 deficiency leads to accumulation of RIG-I protein in human cells and in mice in vivo. As a consequence, Mll5-deficient mice produced more type I IFNs and proinflammatory cytokines in response to RNA virus infection and exhibited enhanced innate immune responses, and reduced viral load and morbidity in vivo.” Whether somewhat higher levels of RIG-I protein in the cell types tested (MEFs, BMDMs, HEK293T) completely or at least in part explain the observed improved antiviral immune response is formally not proven by the presented experiments, although the data are certainly fully compatible with this intuitively attractive interpretation. However, alternative explanations are possible. For instance recent publications have demonstrated that DNA damage can induce a marked IFN response resulting in strong viral resistance of cells [Pepin G et al. NAR 2016, PMID: 27694309] as well as mice [Härtlova A et al. Immunity 2015, PMID: 25692705]. Mll5-deficiency also results in accrued DNA damage and an exaggerated IFN-1 response, as shown recently by Tasdogan et al. Whether DNA damage-mediated antiviral states always require RIG-I stabilization or are due to other, more complex mechanisms, which might also be operative in the system studied here, has not yet been investigated.

A formal demonstration of a causal relationship would require reduction of RIG-I protein levels in Mll5-deficient mice and subsequent demonstration of a correspondingly weaker anti-viral immune defense. Such an experiment could probably be done via introduction of a heterozygous DDX58 mutation onto an Mll5-deficient background (intercross of Mll5-deficient with heterozygous RIG-I-deficient mice). Before we know the result of such an experiment, the authors should definitely formulate much more cautiously.

3. THE AUTHORS FAIL TO ACKNOWLEDGE A PREVIOUSLY DESCRIBED CONNECTION BETWEEN MLL5-DEFICIENCY AND A PHYSIOLOGICALLY RELEVANT ELEVATION OF INFLAMMATORY CYTOKINES: Tasdogan et al. (reference 18 in the submitted manuscript) have dedicated a entire chapter in the

“Results” section to document elevated type 1 IFN levels and their detrimental effects on hematopoietic stem cell function in MLL5-deficient mice. I am surprised that these previous findings are not at all mentioned in support of the data presented now. The authors should correct this omission during revision to make the reader aware of these supportive previous findings. Even if Zhong and colleagues are not the first to highlight a link between MLL5-deficiency and elevated interferon responses, their manuscript doesn’t lose any of its significance and novelty.

Comments/Concerns – Minor:

1. Does the protein band detected with the rabbit antiserum AP14173a (e.g. in figures 4b and c; in figure 5c; in figure 7a) correspond to full-length MLL5 (1868 Aa) or to the major splice variant (908 Aa) or to some other size variant? Please specify throughout the text the MLL5 variant that is meant.
2. Quantitative differences in protein blots are not always as convincing as described in the text (e.g. figure 7e, anti-RIG-I). The authors should include numbers for relative band intensities, whenever the amount of an endogenous protein is determined by WESTERN blotting - as shown in figure 4e. Furthermore, rather than showing just one representative blot, authors should provide quantification data for endogenous proteins on all blots performed within a given experimental set-up, either next to the representative blot or in the supplement. Otherwise statistical evaluation of the results is impossible.
3. Figure 4a: There is no effect on ISRE promotor activity upon coexpression of MLL5 with MAVS, TBK1 or IRF3. The authors should provide evidence that these proteins are indeed expressed in transfected HEK293T cells (e.g. with corresponding WESTERN blots).
4. While the data are generally well presented, the corresponding figures are much too small to be readable without magnification. Figure 5 (page 42) is an extreme example. If the authors want the reader to appreciate their primary data without annoyance, they should increase the size of the figures and the labeling so that all content can be read without magnification glass when printed out on A4-sized paper.

Reviewer #2 (Remarks to the Author):

Summary

In this manuscript the authors report a role of MLL5 (thus far only recognized as an epigenetic modifying factor) in the negative regulation of the innate immune response. Moreover, their data indicate that MLL5 may do so mechanistically by bridging the ubiquitin E3 ligase CHIP with the 5'-ppp RNA sensor RIG-I, thereby mediating RIG-I K48 ubiquitination and its degradation.

- a) BMDMs and PMs from Mll5^{-/-} mice have 2-3 fold more Ifnb, Tnfa, and Il6 mRNA levels upon stimulations that would activate RIG-I. MEFs from these mice have also increased p-IRF3 and p-p65 levels upon VSV infection.
- b) Over-expression of MLL5 inhibits IFN β and NF κ B reporters.
- c) VSV-GFP infection in Mll5^{-/-} mice results in more cytokine production, lower virus titers, increased survival, and decreased lung infiltrates.
- d) Knock-out of Mll5 in HEK-293Ts and BMDMs decreased RIG-I ubiquitination, and increased RIG-I expression and half-life.
- e) Co-IP experiments between CHIP, MLL5 and RIG-I (and mutants) indicated they interact and identified interaction domains.
- f) Co-IPs from Mll5^{-/-} cells and MLL5 over-expressing cells are presented with the claim that MLL5 is essential for the CHIP-RIG-I interaction.
- g) Forced nuclear localization of MLL5 by NLS fusion suggested that the cytoplasmic MLL5 pool is predominantly responsible for RIG-I degradation by CHIP.

Overall reception

Overall, the authors present a body of work from mice and cell culture using both knock-out and overexpression settings supporting their claims. To my knowledge none of the reported findings have been previously reported, and are of interest for a reasonably general audience. The notion that MLL5 had thus far been considered a nuclear protein facilitating epigenetic modification makes it surprising and interesting that it plays a role in the cytoplasm regulating an innate immune sensor.

General comments:

1) This reviewer has one main comment as to whether the claims are supported by the data: although the authors make good individual cases that Mll5^{-/-} mice are more susceptible to VSV infection, and MLL5 affects RIG-I stability, I could not find any experiments that support the claim that these observations are causally related: i.e. that Mll5^{-/-} cells and mice are less susceptible as a result of more RIG-I. In my opinion, an epistasis experiment with VSV infection in RIG-I/MLL5 double and single knock-out 293Ts/MEFs should clarify whether the observed decrease in viral replication is dependent on RIG-I.

Specific comments:

2) Figure 2C: the fold changes in reporter activation by MLL5 over-expression are rather small and occur also in the wildtype cells. Also, MLL5 expression in the knockouts does not rescue to the wild-type levels, and there is no control with polyIC, but without MLL5 expression. It would be important to include this control and show western blots for overexpressed MLL5 in the WT vs KO cells to determine whether any of the differences come from expression level differences.

3) Figure 6A: The authors should include a no-bait control for the IPs (cells without flag-RIGI); it is unclear what is specifically and non-specifically IPed here.

4) In many cases the SEM is plotted and displayed, while in my opinion the SD would be more appropriate.

5) Figure 7a: the authors claim that the cytoplasmic MLL5 increases, while the nuclear pool decreases. Especially the latter statement is difficult to assess. A ratio of the quantification of cytoplasmic MLL5 over actin, and nuclear MLL5 over LaminB should be included as e.g. a bar graph.

6) Supplementary fig 1: for as far as this reviewer can tell the first and only protein level evidence that the Mll5^{-/-} mice do not produce any MLL5 protein (e.g. from secondary start codons, etc) is in figure 4C for BMDMs. In my opinion, it would be important to show evidence for this for MEFs (missing in figure 2), and BMDMs, already when introducing the mouse.

We thank the reviewers for their excellent questions and suggestions, those comments are valuable and helpful for revising our paper. We do our best to answer the questions, the point-by-point responses to the comments are attached below.

Point-by-point Responses to Comments from the Reviewers

Reviewers' comments are *italicized* and our responses are in normal text.

Reviewer #1:

In their study, Zhang et al. provide an exciting set of novel data establishing a surprising connection between the deficiency of MLL5 - a presumed epigenetic regulator - and an enhanced anti-viral immune response. In particular, the authors demonstrate that Mll5-deficient mice newly generated for the study by CRISPR/Cas technology (3 independent lines of Mll5-deficient mice have been described previously by others – references 15 -17 in the manuscript) and cell lines derived thereof exhibit elevated levels of IFNbeta, TNFalpha and IL6 cytokines when infected with RNA, but not DNA viruses. Subsequent in vivo studies reveal that Mll5-deficient mice are markedly more resistant to vesicular stomatitis virus (VSV) infection when compared with wildtype controls. In an extended series of well-controlled biochemical experiments with two primary cell types (MEFs, BMDMs) derived from Mll5 knock-out mice and human HEK293T cells (made Mll5-deficient via CRISPR/Cas), the authors provide conclusive evidence that cytoplasmic MLL5 protein destabilizes the intracellular pattern recognition receptor RIG-I by recruiting the deubiquitinase STUB1, which boosts K48 ubiquitination and proteasomal degradation of RIG-I. The authors propose that the observed stabilization of RIG-I in the absence of Mll5 provides the key mechanism to explain the increased VSV resistance of Mll5-/- mice, as documented here for the first time. The topic investigated and the results obtained are of high scientific interest and should stimulate researchers from distinct fields of investigation. The experiments seem very well performed and the results are convincing. If the authors can address the few concerns mentioned below, I strongly recommend publication.

Major concerns:

C1

INSUFFICIENT CHARACTERIZATION OF NEWLY GENERATED MLL5 MOUSE MUTANTS: Mll5-mutant mice were newly generated for this study by deleting 8 nucleotides in exon 3 using CRISPR/Cas9 technology. The authors state in the “Methods” section: The genotype “of the Mll5-deficient mice was confirmed by sequencing of the PCR fragments in the gRNA-targeting region amplified from genomic DNA isolated from tail tips using the following primers: ...”. No experimental data documenting the effect of this rather limited mutation on MLL5 protein expression are provided. The mouse Mll5 gene is composed of at least 25 coding exons giving rise to a full-length protein of 1868 amino acids. Without data on protein expression, it cannot be excluded that disruption of exon 3 is partially neutralized by in-frame splicing from exon 2 to exon 6, 7 or 8, which would give rise

to a somewhat truncated MLL5 protein of significant length lacking the PHD domain, but still harboring the SET domain. The authors should use antisera directed against the amino- and/or carboxy-terminus of mouse MLL5 to quantify the effect of the CRISPR/Cas9-induced exon 3 mutation on overall protein expression. Antibodies directed against epitopes encoded by exons 3, 4 or 5, like the antiserum (AP14173) used in this study, are clearly not appropriate to document absence of truncated MLL5 protein variants.

Moreover, the authors should provide data documenting to what extent the phenotype of their newly generated Mll5-mutant mice corresponds to the consensus phenotype of the three separate Mll5-deficient mouse strains published previously (references 15 – 17 in the manuscript) - phenotypic features like mild growth retardation, male sterility, strongly reduced thymocyte and splenocyte numbers, radiation sensitivity etc. Without a more comprehensive characterization of the newly generated Mll5-mutant strain the possibility remains that exon 3 disruption has produced either a hypomorph or a truncated protein with dominant negative activity. Similar arguments could be made for the MLL5-mutated HEK293T cell line used for most biochemical studies. The authors should also state how many times the original exon 3 mutation has been back-crossed onto C57BL/6 to minimize potential effects of off-target mutations on the observed phenotype. In case first-generation mouse mutants were used for experiments, the potential problem with off-target mutations in this study should be briefly debated in the “Discussion” section.

R1

*Mll5-deficient mice newly generated in this study exhibited perinatal lethality, postnatal growth retardation, impaired male fertility, and compromised hematopoietic reconstitution, which is consistent with previous reports (*Blood*, 113:1432-1443, 1444-1454, 1455-1463). As shown in **Supplementary Fig. 1d**, the frequency of *Mll5*-deficient mice from *Mll5* heterozygous intercrossing was markedly reduced at weaning age. Surviving *Mll5*-deficient mice showed mild growth retardation (**Supplementary Fig. 1e**). Moreover, *Mll5*-deficient BM cells failed to reconstitute lethally irradiated recipients, confirming that *Mll5* has a critical role in hematopoiesis (**Supplementary Fig. 1f**). All *Mll5*-deficient mice used in this study were backcrossed into C57BL/6 background for at least four generations.*

We tried several commercial anti-MLL5 antibodies (SC-68635, Santa Cruz Biotechnology; SC-68636, Santa Cruz biotechnology; Ab75339, Abcam; SAB1401768, Sigma; AP14173a, Abgent), and found that AP14173a was suitable for the detection of endogenous MLL5 protein. No MLL5 protein was detected in the *MLL5*-deficient HEK293T or MEF cells derived from *Mll5*-deficient mice by using this antibody. To determine further if the translation of MLL5 protein was blocked in *Mll5*^{-/-} mice, we performed RT-PCR and sequence analysis. We found that a premature stop codon (TGA) was induced in exon 3 in *Mll5*-deficient mice (**Supplementary Fig. 1c**). Based on these results, we think that translation of endogenous Mll5 protein was blocked by inducing the 8-bp indel deletion. Nevertheless, we agree with the reviewer that further antisera against the N- or C-terminus of MLL5 protein should be generated to rule out completely the possibility that a truncated dominant negative MLL5 protein caused the phenotypes in our *Mll5*-deficient mice.

C2

THE DATA DO NOT FORMALLY ESTABLISH A CAUSAL RELATIONSHIP BETWEEN ELEVATED RIG-I PROTEIN LEVELS AND THE IMPROVED DEFENSE OF MLL5-DEFICIENT MICE AND CELLS AGAINST RNA VIRUSES: In line with the overall “tone” of the manuscript, the authors state in the “Discussion” (page 18, line 352) “MLL5 deficiency leads to accumulation of RIG-I protein in human cells and in mice in vivo. As a consequence, Mll5-deficient mice produced more type I IFNs and proinflammatory cytokines in response to RNA virus infection and exhibited enhanced innate immune responses, and reduced viral load and morbidity in vivo.” Whether somewhat higher levels of RIG-I protein in the cell types tested (MEFs, BMDMs, HEK293T) completely or at least in part explain the observed improved antiviral immune response is formally not proven by the presented experiments, although the data are certainly fully compatible with this intuitively attractive interpretation. However, alternative explanations are possible. For instance recent publications have demonstrated that DNA damage can induce a marked IFN response resulting in strong viral resistance of cells [Pepin G et al. NAR 2016, PMID: 27694309] as well as mice [Härtlova A et al. Immunity 2015, PMID: 25692705]. Mll5-deficiency also results in accrued DNA damage and an exaggerated IFN-1 response, as shown recently by Tasdogan et al. Whether DNA damage-mediated antiviral states always require RIG-I stabilization or are due to other, more complex mechanisms, which might also be operative in the system studied here, has not yet been investigated.

A formal demonstration of a causal relationship would require reduction of RIG-I protein levels in Mll5-deficient mice and subsequent demonstration of a correspondingly weaker anti-viral immune defense. Such an experiment could probably be done via introduction of a heterozygous DDX58 mutation onto an Mll5-deficient background (intercross of Mll5-deficient with heterozygous RIG-I-deficient mice). Before we know the result of such an experiment, the authors should definitely formulate much more cautiously.

R2

We obtained *Ddx58* gene knockout mice from Prof. Shizuo Akira’s laboratory. Most *Ddx58*-deficient mice died at the embryonic stage, and the remainder died before weaning (*Immunity*. 23, 19-28). Unfortunately, we failed to obtain any surviving *Mll5*, *Ddx58* double knockout mice or *Mll5*-deficient, *Ddx58*-heterozygous mice by intercrossing *Ddx58* heterozygous mice with *Mll5* heterozygous mice. We therefore knocked down RIG-I through sh-RNAs in WT and *Mll5*^{-/-} MEFs (**Fig. 4d**), and challenged them with VSV-GFP and HSV-1, followed by detection of IFN-β and TNF-α expression using qRT-PCR. We found that RIG-I knockdown abrogated augmented expression of *Ifn-β* and *Tnf-α* in *Mll5*^{-/-} MEFs stimulated with VSV-GFP but not HSV-1 (**Fig. 4e**). These results demonstrate that elevated induction of type I IFN and pro-inflammatory cytokines in *Mll5*-deficient cells upon VSV infection were largely dependent on RIG-I.

DNA damage induces type I IFN through the cytosolic DNA sensor STING to promote innate antimicrobial immunity (*Immunity*, 42:332-343.). Tasdogan et al. have demonstrated that *Mll5* deficiency results in accumulation of DNA damage and ROS in HSPCs, and the DNA damage can trigger IFN-1 response and lead to the proliferation and malfunction of HSPCs by the IFN-1/Bid/ROS pathway (*Cell Stem*

Cell, 19:1-16). The levels of type I IFNs in the sera and bone marrow supernatants are comparable between WT and *Mll5*-deficient mice (*Cell Stem Cell*, 19:1-16; **Fig. 5a**). Similarly, no significant changes in IFN- β and TNF- α levels in the sera were detected in our *Mll5*-deficient mice compared to WT mice (**Supplementary Fig. 6**). Therefore, accumulation of RIG-I protein in *Mll5*-deficient cells and resistance to VSV infection of *Mll5*-deficient mice seems unlikely to be due to DNA-damage-induced type I IFN responses. Moreover, *Mll5* deficiency affects induction of IFN- β and TNF- α only upon VSV infection or poly (I:C) transfection, but not HSV-1 infection, or stimulation with other TLR ligands (**Fig. 1a-c**). Therefore, these data indicated that *Mll5*-deficiency-induced augmentation of IFN- β and TNF- α expression upon RNA virus infection is mainly dependent on enhanced RIG-I stability.

C3

THE AUTHORS FAIL TO ACKNOWLEDGE A PREVIOUSLY DESCRIBED CONNECTION BETWEEN MLL5-DEFICIENCY AND A PHYSIOLOGICALLY RELEVANT ELEVATION OF INFLAMMATORY CYTOKINES: Tasdogan et al. (reference 18 in the submitted manuscript) have dedicated a entire chapter in the "Results" section to document elevated type I IFN levels and their detrimental effects on hematopoietic stem cell function in Mll5-deficient mice. I am surprised that these previous findings are not at all mentioned in support of the data presented now. The authors should correct this omission during revision to make the reader aware of these supportive previous findings. Even if Zhang and colleagues are not the first to highlight a link between Mll5-deficiency and elevated interferon responses, their manuscript doesn't lose any of its significance and novelty.

R3

Tasdogan *et al.* have demonstrated that *Mll5* deficiency causes DNA damage and associated IFN-1 signaling in mitochondrial Bid mobilization. We have discussed their findings in the revised manuscript.

Minor concerns:

C1

Does the protein band detected with the rabbit antiserum AP14173a (e.g. in figures 4b and c; in figure 5c; in figure 7a) correspond to full-length MLL5 (1868 Aa) or to the major splice variant (908 Aa) or to some other size variant? Please specify throughout the text the MLL5 variant that is meant.

R1

In addition to the full-length MLL5, another three isoforms (MLL5 α , MLL5 β and NKp44L) are documented (reference listed below). The full-length MLL5 contains 25 coding exons, which produce a protein of 1858 amino acids. MLL5 α is translated to a protein of 608 amino acids, which is from exons 1 to 13. We found that the antibody AP14173a was suitable for the detection of endogenous MLL5 α isoform. However, we failed to detect the full-length MLL5 protein using this antibody.

Reference:

Zhang, X., Novera, W., Zhang, Y. & Deng, L.-W. MLL5 (KMT2E): structure, function, and clinical relevance. *Cellular and Molecular Life Sciences*, doi:10.1007/s00018-017-2470-8 (2017).

C2

Quantitative differences in protein blots are not always as convincing as described in the text (e.g. figure 7e, anti-RIG-I). The authors should include numbers for relative band intensities, whenever the amount of an endogenous protein is determined by WESTERN blotting - as shown in figure 4e. Furthermore, rather than showing just one representative blot, authors should provide quantification data for endogenous proteins on all blots performed within a given experimental set-up, either next to the representative blot or in the supplement. Otherwise statistical evaluation of the results is impossible.

R2

In the revised manuscript, relative band densities (the ratio of the quantification of RIG-I over actin) indicating protein levels are shown below each band (**Fig. 7e**). Extra data for quantification of endogenous RIG-I protein levels are shown in **supplementary Table 3**. The band density indicating protein concentration was quantified with Image J software.

C3

Figure 4a: There is no effect on ISRE promotor activity upon co-expression of MLL5 with MAVS, TBK1 or IRF3. The authors should provide evidence that these proteins are indeed expressed in transfected HEK293T cells (e.g. with corresponding WESTERN blots).

R3

In the revised manuscript, we showed the western blots for overexpressed MLL5 with MAVS, TBK1 or IRF3 in the WT 293T cells.

C4

While the data are generally well presented, the corresponding figures are much too small to be readable without magnification. Figure 5 (page 42) is an extreme example. If the authors want the reader to appreciate their primary data without annoyance, they should increase the size of the figures and the labeling so that all content can be read without magnification glass when printed out on A4-sized paper.

R4

In the revised manuscript, we increased the size of the figures and the labeling.

Reviewer #2:

In this manuscript the authors report a role of MLL5 (thus far only recognized as an epigenetic modifying factor) in the negative regulation of the innate immune response. Moreover, their data indicate that MLL5 may do so mechanistically by bridging the ubiquitin E3 ligase CHIP with the 5'-ppp RNA sensor RIG-I, thereby mediating RIG-I K48 ubiquitination and its degradation.

a) BMDMs and PMs from Mll5^{-/-} mice have 2-3 fold more Ifnb, Tnfa, and Il6 mRNA levels upon stimulations that would activate RIG-I. MEFs from these mice have also increased p-IRF3 and p-p65 levels upon VSV infection.

b) Over-expression of MLL5 inhibits IFN β and NF κ B reporters.

c) VSV-GFP infection in Mll5^{-/-} mice results in more cytokine production, lower virus titers, increased survival, and decreased lung infiltrates.

d) Knock-out of Mll5 in HEK-293Ts and BMDMs decreased RIG-I ubiquitination, and increased RIG-I expression and hal-life.

e) Co-IP experiments between CHIP, MLL5 and RIG-I (and mutants) indicated they interact and identified interaction domains.

f) Co-IPs from Mll5^{-/-} cells and MLL5 over-expressing cells are presented with the claim that MLL5 is essential for the CHIP-RIG-I interaction.

g) Forced nuclear localization of MLL5 by NLS fusion suggested that the cytoplasmic MLL5 pool is predominantly responsible for RIG-I degradation by CHIP.

Overall reception

Overall, the authors present a body of work from mice and cell culture using both knock-out and overexpression settings supporting their claims. To my knowledge none of the reported findings have been previously reported, and are of interest for a reasonably general audience. The notion that MLL5 had thus far been considered a nuclear protein facilitating epigenetic modification makes it surprising and interesting that it plays a role in the cytoplasm regulating an innate immune sensor.

General comments:**C1**

This reviewer has one main comment as to whether the claims are supported by the data: although the authors make good individual cases that Mll5^{-/-} mice are more susceptible to VSV infection, and MLL5 affects RIG-I stability, I could not find any experiments that support the claim that these observations are causally related: i.e. that Mll5^{-/-} cells and mice are less susceptible as a result of more RIG-I. In my opinion, an epistasis experiment with VSV infection in RIG-I/MLL5 double and single knock-out 293Ts/MEFs should clarify whether the observed decrease in viral replication is dependent on RIG-I.

R1

We obtained Ddx58 gene knockout mice from Prof. Shizuo Akira's laboratory. Most Ddx58-deficient mice died at the embryonic stage, and the remainder died before

weaning (*Immunity* 23, 19-28). Unfortunately, we failed to obtain any surviving *Mll5*, *Ddx58* double knockout mice or *Mll5*-deficient, *Ddx58*-heterozygous mice by intercrossing *Ddx58*-heterozygous mice with *Mll5*-heterozygous mice. We therefore knocked down RIG-I through sh-RNAs in WT and *Mll5*^{-/-} MEFs (**Fig. 4d**), and challenged them with VSV-GFP and HSV-1, followed by detection of IFN- β and TNF- α expression using qRT-PCR. We found that RIG-I knockdown abrogated augmented expression of *Ifn- β* and *Tnf- α* in *Mll5*^{-/-} MEFs stimulated with VSV-GFP but not HSV-1 (**Fig. 4e**). These results demonstrate that elevated induction of type I IFN and pro-inflammatory cytokines in *Mll5*-deficient cells upon VSV infection largely relied upon RIG-I.

Specific comments:

C1

Figure 2C: the fold changes in reporter activation by MLL5 over-expression are rather small and occur also in the wildtype cells. Also, MLL5 expression in the knockouts does not rescue to the wild-type levels, and there is no control with polyIC, but without MLL5 expression. It would be important to include this control and show western blots for overexpressed MLL5 in the WT vs KO cells to determine whether any of the differences come from expression level differences.

R1

In the revised manuscript, we showed the western blots for overexpressed MLL5 in the WT and KO HEK293T cells. For the control with poly(I:C) but without MLL5 overexpression, the second bars of each group were the results with poly(I:C) but without MLL5 overexpression. We made it clear in the revised manuscript.

C2

Figure 6A: The authors should include a no-bait control for the IPs (cells without flag-RIGI); it is unclear what is specifically and non-specifically IPed here.

R2

We performed the experiments with a no-bait control for the IPs (cells without FLAG-RIG-I knock-in); the new results are showed in **Fig. 6A**.

C3

In many cases the SEM is plotted and displayed, while in my opinion the SD would be more appropriate.

R3

In the revised manuscript, the results are presented as mean \pm SD.

C4

Figure 7a: the authors claim that the cytoplasmic MLL5 increases, while the nuclear pool decreases. Especially the latter statement is difficult to assess. A ratio of the quantification of cytoplasmic MLL5 over actin, and nuclear MLL5 over LaminB should be included as e.g. a bar graph.

R4

In the revised manuscript, relative band densities (the ratio of the quantification of cytoplasmic MLL5 over actin, and nuclear MLL5 over Lamin B) indicating protein levels are shown below each band. The extra data for quantification of endogenous MLL5 protein levels are shown in **Supplementary Table 2**. The band density indicating protein amount was quantified with Image J software.

C5

*Supplementary fig 1: for as far as this reviewer can tell the first and only protein level evidence that the *Mll5*^{-/-} mice do not produce any MLL5 protein (e.g. from secondary start codons, etc) is in figure 4C for BMDMs. In my opinion, it would be important to show evidence for this for MEFs (missing in figure 2), and BMDMs, already when introducing the mouse.*

R5

In the revised manuscript, we showed the absence of MLL5 protein in BMDMs and MEFs derived from *Mll5*^{-/-} mice (**Fig. 4 c, d**). Moreover, we performed RT-PCR and sequence analysis in *Mll5*^{-/-} MEF cells, and found that a premature stop codon (TGA) was induced in exon 3 in *Mll5*-deficient mice (**Supplementary Fig. 1c**).

Mll5-deficient mice newly generated in this study exhibited perinatal lethality, postnatal growth retardation, impaired male fertility, and compromised hematopoietic reconstitution, which is consistent with previous reports (*Blood*, 113:1432-1443, 1444-1454, 1455-1463). The frequency of *Mll5*-deficient mice from *Mll5* heterozygous intercrossing was markedly reduced at weaning age (**Supplementary Fig. 1d**). Surviving *Mll5*-deficient mice showed mild growth retardation (**Supplementary Fig. 1e**). Moreover, *Mll5*-deficient BM cells failed to reconstitute lethally irradiated recipients, confirming that *Mll5* has a critical role in hematopoiesis (**Supplementary Fig. 1f**). All *Mll5*-deficient mice used in this study were backcrossed into C57BL/6 background for at least four generations.

REVIEWERS' COMMENTS:

Reviewer #1 (Remarks to the Author):

In my previous review, I strongly recommended publication of the manuscript, provided the authors (Zhong et al.) would be able to clear three major and four minor concerns. In the revised version of the manuscript, the authors have made substantial efforts to address these issues, as specified below for each point:

My original concerns and how they have been addressed – Major:

1. Insufficient characterization of newly generated Mll5 mouse mutants:

1.1 In new Supplementary Figure 1c the authors document the occurrence of a stop codon in exon 3 ensuing CRISPR/Cas-mediated gene mutation as evidence for a complete knock-out. The authors point out that currently no appropriate antibody reagents (antisera or monoclonals) are available to completely rule out the generation of truncated MLL5 versions in their homozygous knock-out strain.

My new comments to point 1.1:

I agree with the authors that new Supplementary Fig.1c in combination with a description of the overall phenotype of newly generated Mll5-deficient mice (new Supplementary Fig.1d-f) document beyond reasonable doubt that the CRISPR/Cas-mediated mutation indeed results in a complete Mll5 gene knock-out. However, the authors should not forget to refer to new Supplementary Fig.1c in the main text.

1.2 Does the overall phenotype of the newly generated strain of Mll5-deficient mice correspond to previously published phenotypes? In the revised version, the authors have added Supplementary Fig. 1c, d, and e, documenting partial embryonic (or perinatal) lethality (S1c) and growth retardation (1d) among homozygous Mll5^{-/-} mice, as well as impaired reconstitution ability of Mll5-deficient BM cells (1d). In their response to the reviewer, the authors write: >> Mll5-deficient mice newly generated in this study exhibited perinatal lethality, postnatal growth retardation, impaired male fertility, and compromised hematopoietic reconstitution, which is consistent with previous reports (Blood, 113:1432-1443, 1444-1454, 1455-1463)<<.

My new comments to point 1.2:

The authors should include this sentence (bracketed by >> <<) in the main text and should also not forget to refer to the new figures S1d-f in the main text.

All in all, my major concern #1 has been addressed satisfactorily.

2. My original concern: The data do not formally establish a causal relationship between elevated RIG-I protein levels and the improved defense of Mll5-deficient mice/cells against RNA viruses.

The authors report that their efforts to formally establish a causal relationship between elevated RIG-I protein levels and the improved defense of Mll5-deficient mice against RNA viruses by reducing RIG-I protein levels genetically failed as adult Mll5/Ddx58 (encoding mouse RIG-I)-double deficient mice could not be obtained (embryonic lethality?). To demonstrate a causal relationship at least in cells cultured in vitro, the authors provide as further evidence confirmatory results of a RIG-I knock-down experiment performed with WT and Mll5^{-/-} MEFs (new Fig.4d and 4e).

My new comments to point 2:

Confirmatory results in mouse embryonic fibroblasts (MEFs), as encouraging as they may be, also fail to formally prove that the mechanism described in this manuscript (fully) explains the resistance of Mll5-deficient mice towards RNA viruses. To recall, all details of the described mechanism (ubiquitination and degradation of RIG-I by MLL5-mediated recruitment of STUB1) were elucidated either in MEFs or HEK293 cells. Obviously, neither the former nor the latter cells (of human embryonic kidney origin) are involved in adult mice in any kind of immune response, also not against RNA viruses. Although the results obtained with MEFs and HEK293 cells provide an exciting and plausible scenario of how Mll5-deficiency may improve viral resistance in the intact organism, this new and attractive hypothesis remains formally unproven. Sentences like >> These results explain why Mll5-deficient mice produced more type I IFNs and exhibited enhanced antiviral innate immune responses upon RNA virus infection. << (pages 18/19, lines 382 – 384) are thus scientifically incorrect and should be modified at least by inserting a cautious “may”. To get me right: I am not asking the authors to provide the missing formal experimental evidence at this stage, as this is anything but trivial and obviously would require a whole set of additional comprehensive in vivo experimentation, probably including additional mouse mutants. The data provided by the authors here are on their own of sufficient quality and interest to merit publication in a high impact journal. However, in all fairness, I must insist that the authors should point out explicitly in 2-3 sentences in the “Discussion” that their findings do not formally establish a causal relationship between the described mechanism and an improved defense of Mll5-deficient mice against RNA viruses at the organismal level. I do not think that my concern is a minor issue. For instance, if the same or other authors after much

additional work may eventually succeed in providing conclusive in vivo evidence for the presumed relationship, there is a clear risk that they will be unjustly confronted by referees or journal editors that their findings are not truly novel – a possible outcome because of vastly exaggerated statements in previous publications.

3. Original concern: The authors fail to acknowledge a previously described connection between MII5-deficiency and a physiologically relevant elevation of type I interferons.

In the revised manuscript, the authors comment on this issue adequately in the “Discussion” section (page 19, lanes 385 – 401).

My minor comments/concerns #1 - #3 have been adequately addressed in the revised version of the manuscript. Unfortunately, also after revision, the size of many figures along with the corresponding legends is so small that, when printed, important experimental details cannot be discerned without magnification glasses (my minor point #4). I find this quite annoying. Of course, in the end, it's up to the journal editors to decide, whether provided figure formats are acceptable.

New minor concerns/comments:

1. The statement >> The total protein levels of RIG-I in wild-type HEK293T cells dropped below 30% at 30 h after CHX treatment << (page 11, lane 220/221) is not in agreement with the values given in Supplementary Table 1 (60% and 47% for experiments 1 and 2, respectively).
2. The * indicating IgG light chain, as mentioned in the legend, is missing in figure 6B.

Reviewer #2 (Remarks to the Author):

Review revised manuscript NCOMMS-17-21397A

Reviewer #2

The authors addressed my main concern (C1.1; whether the effect of MII5 ablation on the antiviral response was RIG-I-dependent) by a convincing epistasis experiment.

Additional experiments were carried out to clarify some missing controls (C1.2, C2, C5); all of these support the author's claims and conclusions. Lastly, two additional minor points (C3, C4) were appropriately addressed by quantification and adjusting statistical analyses.

The authors have in my opinion very well addressed all of my previous comments with new experimental data. I do not have any further remarks besides congratulations on a very interesting and thorough body of work.

We thank the reviewers for their excellent comments and suggestions, those comments are all valuable and very helpful for revising and improving our paper. We answer their questions, the point-by-point responses to the comments are attached below.

REVIEWERS' COMMENTS:

Reviewer #1 (Remarks to the Author):

In my previous review, I strongly recommended publication of the manuscript, provided the authors (Zhong et al.) would be able to clear three major and four minor concerns. In the revised version of the manuscript, the authors have made substantial efforts to address these issues, as specified below for each point:

My original concerns and how they have been addressed – Major:

1. Insufficient characterization of newly generated Mll5 mouse mutants:

1.1 In new Supplementary Figure 1c the authors document the occurrence of a stop codon in exon 3 ensuing CRISPR/Cas-mediated gene mutation as evidence for a complete knock-out. The authors point out that currently no appropriate antibody reagents (antisera or monoclonals) are available to completely rule out the generation of truncated MLL5 versions in their homozygous knock-out strain.

My new comments to point 1.1:

I agree with the authors that new Supplementary Fig.1c in combination with a description of the overall phenotype of newly generated Mll5-deficient mice (new Supplementary Fig.1d-f) document beyond reasonable doubt that the CRISPR/Cas-mediated mutation indeed results in a complete Mll5 gene knock-out. However, the authors should not forget to refer to new Supplementary Fig.1c in the main text.

R1.1:

We thank the reviewer for raising the issue. We described the overall phenotypes of newly generated Mll5-deficient mice (**Supplementary Figure 1**) in the revised manuscript (line93-98).

1.2 Does the overall phenotype of the newly generated strain of Mll5-deficient mice correspond to previously published phenotypes? In the revised version, the authors have added Supplementary Fig. 1c, d, and e, documenting partial embryonic (or perinatal) lethality (S1c) and growth retardation (1d) among homozygous Mll5-/- mice, as well as impaired reconstitution ability of Mll5-deficient BM cells (1d). In their response to the reviewer, the authors write: >> Mll5-deficient mice newly generated in this study exhibited perinatal lethality, postnatal growth retardation, impaired male fertility, and compromised hematopoietic reconstitution, which is consistent with previous reports (Blood, 113:1432-1443, 1444-1454, 1455-1463)<<.

My new comments to point 1.2:

The authors should include this sentence (bracketed by >> <<) in the main text and should also not forget to refer to the new figures S1d-f in the main text.

R1.2:

We thank the reviewer for raising this issue. We added the sentence “Mll5-deficient mice newly generated in this study exhibited perinatal lethality, postnatal growth retardation, impaired male fertility, and compromised hematopoietic reconstitution, which is consistent with previous reports” into the main text (Line 93-98).

All in all, my major concern #1 has been addressed satisfactorily.

2. My original concern: *The data do not formally establish a causal relationship between elevated RIG-I protein levels and the improved defense of Mll5-deficient mice/cells against RNA viruses.*

The authors report that their efforts to formally establish a causal relationship between elevated RIG-I protein levels and the improved defense of Mll5-deficient mice against RNA viruses by reducing RIG-I protein levels genetically failed as adult Mll5/Ddx58 (encoding mouse RIG-I)-double deficient mice could not be obtained (embryonic lethality?). To demonstrate a causal relationship at least in cells cultured in vitro, the authors provide as further evidence confirmatory results of a RIG-I knock-down experiment performed with WT and Mll5^{-/-} MEFs (new Fig.4d and 4e).

My new comments to point 2:

Confirmatory results in mouse embryonic fibroblasts (MEFs), as encouraging as they may be, also fail to formally prove that the mechanism described in this manuscript (fully) explains the resistance of Mll5-deficient mice towards RNA viruses. To recall, all details of the described mechanism (ubiquitination and degradation of RIG-I by MLL5-mediated recruitment of STUB1) were elucidated either in MEFs or HEK293 cells. Obviously, neither the former nor the latter cells (of human embryonic kidney origin) are involved in adult mice in any kind of immune response, also not against RNA viruses. Although the results obtained with MEFs and HEK293 cells provide an exciting and plausible scenario of how Mll5-deficiency may improve viral resistance in the intact organism, this new and attractive hypothesis remains formally unproven. Sentences like >> These results explain why Mll5-deficient mice produced more type I IFNs and exhibited enhanced antiviral innate immune responses upon RNA virus infection. << (pages 18/19, lines 382 – 384) are thus scientifically incorrect and should be modified at least by inserting a cautious “may”. To get me right: I am not asking the authors to provide the missing formal experimental evidence at this stage, as this is anything but trivial and obviously would require a whole set of additional comprehensive in vivo experimentation, probably including additional mouse mutants. The data provided by the authors here are on their own of sufficient quality and interest to merit publication in a high impact journal. However, in all fairness, I must insist that the authors should point out explicitly in 2-3 sentences in the “Discussion” that their findings do not formally establish a causal relationship between the described mechanism and an improved defense of Mll5-deficient mice against RNA viruses at the organismal level. I do not think that my concern is a minor issue. For instance, if the same or other authors after much additional work may eventually succeed in providing conclusive in vivo evidence for the presumed relationship, there is a clear risk that they will be unjustly confronted by referees or journal editors that their findings are not truly novel – a possible outcome because of vastly exaggerated statements in previous publications.

R2:

We thank the reviewer for raising this important issue. We inserted a cautious “might” into discussion part (Line379). In addition, we added a sentence “These results further indicated that, *although RIG-I-independent mechanism could not be fully excluded, Mll5-deficiency-induced augment of expression of IFN-β is largely dependent on RIG-I*” into the discussion part (line394-397).

3. Original concern: *The authors fail to acknowledge a previously described connection between Mll5-deficiency and a physiologically relevant elevation of type I interferons. In the revised manuscript, the authors comment on this issue adequately in the “Discussion”*

section (page 19, lanes 385 – 401).

My minor comments/concerns #1 - #3 have been adequately addressed in the revised version of the manuscript. Unfortunately, also after revision, the size of many figures along with the corresponding legends is so small that, when printed, important experimental details cannot be discerned without magnification glasses (my minor point #4). I find this quite annoying. Of course, in the end, it's up to the journal editors to decide, whether provided figure formats are acceptable.

New minor concerns/comments:

1. The statement >> The total protein levels of RIG-I in wild-type HEK293T cells dropped below 30% at 30 h after CHX treatment << (page 11, lane 220/221) is not in agreement with the values given in Supplementary Table 1 (60% and 47% for experiments 1 and 2, respectively).

R1:

We thanks the reviewer for reasing this issue. We changed it to more accurate description in the resubmitted manuscript (line220-221).

*2. The * indicating IgG light chain, as mentioned in the legend, is missing in figure 6B.*

R2:

Thanks the reviewer and we added “*” indicating IgG light chain into **Figure 6B** in the resubmitted manuscript.

Reviewer #2 (Remarks to the Author):

Review revised manuscript NCOMMS-17-21397A

Reviewer #2

The authors addressed my main concern (C1.1; whether the effect of Mll5 ablation on the antiviral response was RIG-I-dependent) by a convincing epistasis experiment. Additional experiments were carried out to clarify some missing controls (C1.2, C2, C5); all of these support the author's claims and conclusions. Lastly, two additional minor points (C3, C4) were appropriately addressed by quantification and adjusting statistical analyses.

The authors have in my opinion very well addressed all of my previous comments with new experimental data. I do not have any further remarks besides congratulations on a very interesting and thorough body of work.

R:

Thank you very much!